# Antibody targeting intracellular oncogenic Ras mutants exerts anti-tumour effects after systemic administration

Seung-Min Shin[1,*], Dong-Ki Choi[1,*,†], Keunok Jung[2], Jeomil Bae[1,†], Ji-sun Kim[1], Seong-wook Park[1], Ki-Hoon Song[3] & Yong-Sung Kim[1]

Oncogenic Ras mutants, frequently detected in human cancers, are high-priority anticancer drug targets. However, direct inhibition of oncogenic Ras mutants with small molecules has been extremely challenging. Here we report the development of a human IgG1 format antibody, RT11, which internalizes into the cytosol of living cells and selectively binds to the activated GTP-bound form of various oncogenic Ras mutants to block the interactions with effector proteins, thereby suppressing downstream signalling and exerting anti-proliferative effects in a variety of tumour cells harbouring oncogenic Ras mutants. When systemically administered, an RT11 variant with an additional tumour-associated integrin binding moiety for tumour tissue targeting significantly inhibits the *in vivo* growth of oncogenic Ras-mutated tumour xenografts in mice, but not wild-type Ras-harbouring tumours. Our results demonstrate the feasibility of developing therapeutic antibodies for direct targeting of cytosolic proteins that are inaccessible using current antibody technology.

[1] Department of Molecular Science and Technology, Ajou University, Suwon 16499, Republic of Korea. [2] Priority Research Center for Molecular Science & Technology, Ajou University, Suwon 16499, Republic of Korea. [3] Department of Allergy and Clinical Immunology, School of Medicine, Ajou University, Suwon 16499, Republic of Korea. * These authors contributed equally to this work. † Present address: ORUM Therapeutics, Inc., 11-3, Techno 1-ro, Yuseong-gu, Daejeon, Republic of Korea (D.K.C); Center for Vascular Research, Institute for Basic Science, Daejeon 34141, Republic of Korea (J.B). Correspondence and requests for materials should be addressed to Y.-S.K. (email: kimys@ajou.ac.kr).

Ras proteins (KRas, HRas and NRas) are small GTPases that function as molecular switches at the inner plasma membrane by alternating between GTP-bound active forms (Ras·GTP) and GDP-bound (Ras·GDP) inactive forms, which differ based on the conformations of their switch I and switch II regions[1,2]. Only active Ras·GTP interacts through its two distinct switch regions with the conserved Ras-binding domain (RBD) of multiple effector proteins, such as Raf kinases, PI3K (phosphatidylinositol 3-kinase) and RalGDS (Ral guanine nucleotide dissociation stimulator)[3,4]. These protein–protein interactions (PPIs) trigger many cellular signalling of proliferation, differentiation and survival[1,2]. Oncogenic mutations in Ras proteins, predominantly found at G12, G13 and Q61 residues, impair the GTPase activity rendering the mutants persistently GTP-bound active form, thereby promoting tumorigenesis and tumour malignancy[1,2]. Oncogenic Ras mutants, and most frequently KRas mutants (86% of Ras-driven cancers)[5], are found in ~25% of human cancers with the highest frequencies in pancreatic (~98%) and colorectal (~53%) cancers[1,6].

Decades of efforts have been made to develop oncogenic Ras mutant-targeting small molecule agents[7–12]. However, direct inhibition of oncogenic Ras mutants has proven extremely challenging and no agents have been clinically approved to date; this is mainly due to difficulties in identifying druggable pockets for small molecule binding on the surface of Ras[13]. Recently, the small-molecule rigosertib, which binds to the RBD of effector proteins rather than Ras itself, was shown to block the PPIs between Ras and effector proteins[14]. Alternatively, some peptide-based inhibitors that block the PPIs between active Ras and effector proteins have been reported[15,16]; however, the in vivo activities were not evaluated.

Antibodies with large surface area paratopes are excellent to specifically target proteins with high affinity[17]. More than 40 therapeutic antibodies have been clinically approved against many extracellular proteins[18]. However, such antibodies do not have the capacity to localize to cellular cytosolic regions after receptor-mediated endocytosis[19], restricting their therapeutic application in targeting cytosolic proteins. Previously, intracellularly expressed antibody fragments (intrabodies) were developed that selectively bind to the active Ras·GTP form to block the PPIs with effector proteins, thereby inhibiting tumorigenesis and metastasis in mouse models[20,21]. This suggests that blocking intracellular Ras·GTP–effector PPIs using a conventional antibody regimen such as systemic administration could be an effective approach to inhibit oncogenic Ras-driven signalling.

Here we describe the generation and therapeutic efficacy of a human IgG1 format antibody, named iMab (internalizing and PPI interfering monoclonal antibody), which directly targets the intracellularly activated GTP-bound form of oncogenic Ras mutants after internalization into the cytosol of living cells. iMab specifically binds to the PPI interfaces between activated Ras and effector proteins to block these associations, thereby inhibiting downstream oncogenic signalling and exerting anti-tumour effects in mouse xenograft models when systemically administered.

## Results

### Generation of GTP-bound active Ras specific RT11 iMab. We recently reported a cytosol-penetrating antibody TMab4, referred to as cytotransmab[22]. In the intact human IgG1 form, this can reach the cytosol of living cells after internalization through clathrin-mediated endocytosis using cell surface-expressed heparan sulfate proteoglycan (HSPG) as a receptor, and

subsequent endosomal escape from early endosomes into the cytosol[22,23]. Internalized TMab4 undergoes conformational changes in response to the acidified pH of early endosomes, which results in endosomal membrane pore formation, through which TMab4 escapes into the cytosol[24]. Since the capacity for cellular internalization and subsequent endosomal escape of TMab4 cytotransmab resides in the light chain variable domain (VL)[22,24], we sought to generate an active Ras·GTP-specific iMab by replacing the original heavy chain variable domain (VH) of TMab4 with a Ras·GTP specific-binding VH domain in the conventional IgG1 format (Fig. 1a). As an antigen to screen such VH domains, the active form of human KRas mutant with a G12D mutation (KRas[G12D]), which is the most prevalent mutant found in Ras-driven tumours[1,5], was prepared by loading it with the non-hydrolysable GTP analog, GppNHp (guanosine 5′-[β,γ-imido] triphosphate). The conformations of GppNHp-bound KRas mutants are equivalent to those of the GTP-bound forms with resistance to γ-phosphate hydrolysis[4,11,20]. To isolate active Ras form–specific VH domains, we screened synthetic human heavy chain (HC) libraries displayed in the format of VH library-CH1 on the surface of yeast haploid cells[25]. This was performed by one round of magnetic activated cell sorting (MACS) and then one round of fluorescence activated cell sorting (FACS) against biotinylated KRas[G12D]·GppNHp antigen in the presence of at least a 10-fold excess molar ratio of non-biotinylated KRas[G12D]·GDP as a soluble competitor (Fig. 1b). The enriched HC yeast haploid library cells were then mated with the other mating type of yeast cells that secret the light chain (LC) with the VL-CL of TMab4 in an attempt to isolate a Ras·GTP specific VH that can properly associate with TMab4 VL in the IgG form (Fig. 1b). Two subsequent rounds of FACS of the mated Fab library in the mode of competition screening generated the RT4 clone. When characterized in the purified human IgG1 form, RT4 selectively bound KRas[G12D]·GppNHp, but not KRas[G12D]·GDP, with a dissociation constant ($K_D$) of ~110 nM (Supplementary Fig. 1a–c).

To further increase the affinity of RT4 for active Ras form, we constructed three types of HC libraries based on the RT4 VH domain in yeast haploid cells (Supplementary Fig. 2a) and screened them against KRas[G12D]·GppNHp in competition mode using KRas[G12D]·GDP with one round of MACS and then mated the enriched library cells with the yeast cells that secret TMab4 LC. The mated diploid yeast cells were further subjected to competition screening by FACS, yielding 6 unique high affinity RT4 variants (Supplementary Fig. 2b). Evaluation of these variants in the purified human IgG1 form for selective binding activity of KRas[G12D]·GppNHp over KRas[G12D]·GDP led us to choose RT11 for further study (Supplementary Fig. 2c–f).

### RT11 binds to the PPI interfaces of active Ras form. RT11 bound selectively to the GppNHp-bound active forms of wild-type (WT) KRas, NRas, and HRas and their oncogenic mutants with mutations at residue 12, 13 or 61, such as KRas[G12D], KRas[G12V], KRas[G13D], KRas[Q61H], HRas[G12V] and NRas[Q61R], but negligibly to the GDP-bound inactive forms (Fig. 1c). This demonstrated that RT11 specifically binds to the activated forms of all three Ras proteins, regardless of isotype and oncogenic mutations. When measured by surface plasmon resonance, the $K_D$ values ranged from 4 to 17 nM (Supplementary Table 1), which suggests a much stronger affinity than that of active Ras for its effector proteins, such as approximately 56–160 nM for Raf[26–28], 1 μM for RalGDS[29] and 2.7 μM for PI3K[30,31]. RT11 efficiently competed with the RBDs of cRaf (cRaf[RBD]) and RalGDS

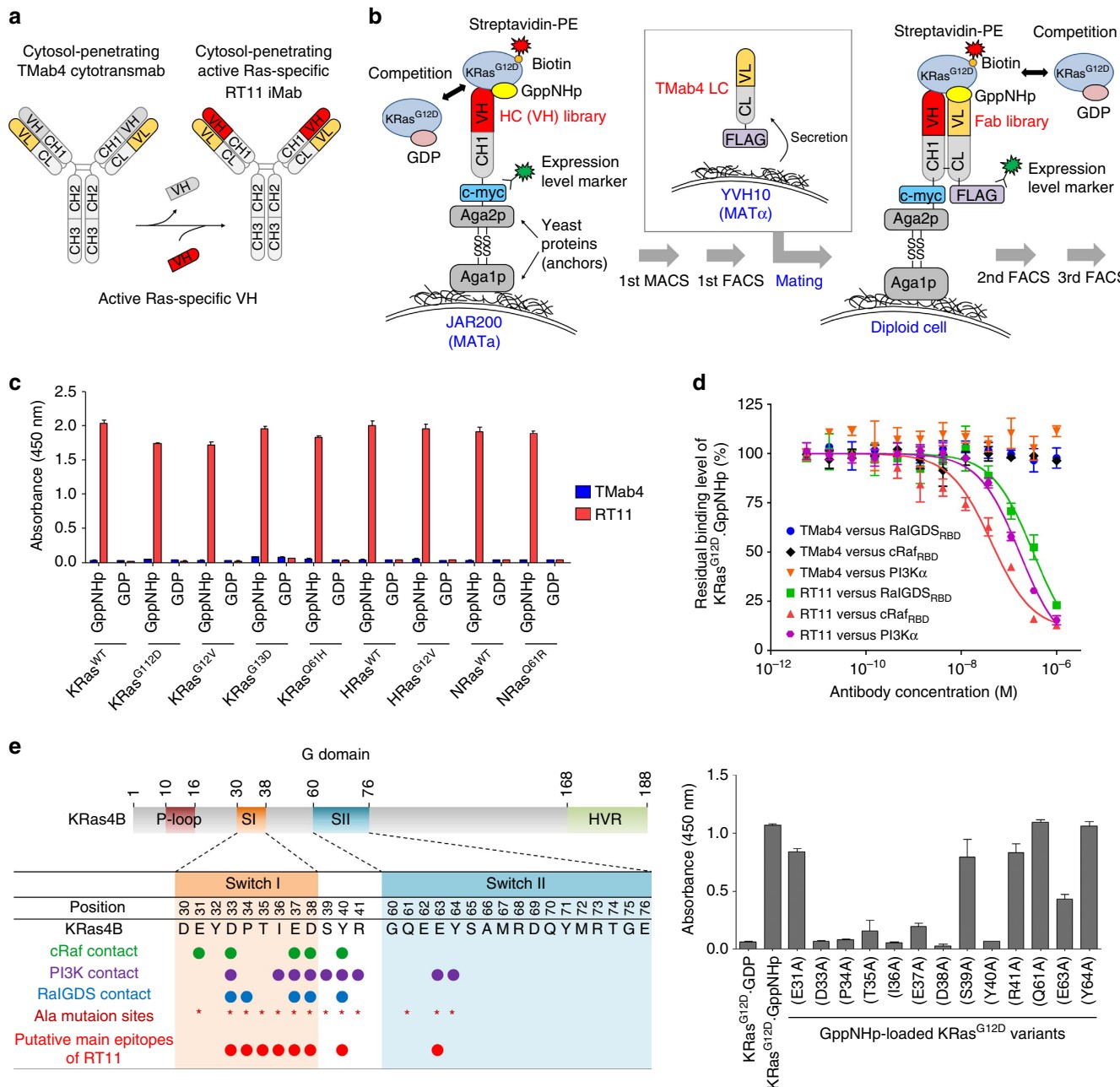

**Figure 1 | Generation and characterization of active Ras specific RT11 iMab.** (**a**) A strategy to generate cytosol-penetrating and GTP-bound active Ras specific antibody, RT11 iMab. (**b**) Schematic of screening method to isolate VH-dependent active Ras form-specific binding iMab antibody. (**c**) ELISA showing the selective binding of RT11, but not TMab4, to the GppNHp-bound active form of the representative oncogenic Ras mutants over the GDP-bound inactive form. (**d**) Competition ELISA showing that RT11, but not TMab4, blocks interactions between $KRas^{G12D} \cdot GppNHp$ and effector proteins, such as $cRaf_{RBD}$, $RalGDS_{RBD}$, and $PI3K\alpha$ (p110$\alpha$/p85$\alpha$). (**e**) Epitope mapping of RT11 by alanine-scanning mutagenesis. (Right) ELISA showing the binding of RT11 to $KRas^{G12D}$ alanine mutants. (Left) Comparison of the KRas4B residues directly involved in the interactions with cRaf[61], PI3K[30], and RalGDS[62] with the putative main epitopes (red circle) of RT11, determined by alanine-scanning mutagenesis. In **c–e**, error bars represent the mean ± s.d. (n = 3).

($RalGDS_{RBD}$), as well as PI3K$\alpha$ (p110$\alpha$/p85$\alpha$), for binding to $KRas^{G12D} \cdot GppNHp$ (Fig. 1d), suggesting that RT11 binds to the PPI interfaces between active Ras and effector proteins.

To further identify the epitopes recognized by RT11, alanine-scanning mutagenesis was performed focusing on residues in direct contact with effector proteins in the switch I (residues 30–38) and switch II (residues 60–76) regions of $KRas^{G12D}$ (Fig. 1e)[3,4,30]. RT11 significantly lost binding affinity for $KRas^{G12D} \cdot GppNHp$ mutants carrying an alanine substitution at most residues in the switch I (D33, P34, T35, I36, E37, D38 and Y40) and switch II regions (E63), with the exception of some residues (E31, S39, R41,

Q61 and Y64; Fig. 1e and Supplementary Fig. 3a). This mutational study, combined with the results of competition ELISA, strongly suggests that RT11 specifically binds to the PPI interfaces, namely the switch I and II regions, of active Ras form with the effector proteins (Fig. 1e and Supplementary Fig. 3b).

When assessed with other Ras family members[32], RT11 exhibited ∼50% binding activity for GppNHp-bound active forms of $MRas^{WT}$ and $RRas^{WT}$, but not for the GDP-bound inactive ones, when compared to active $KRas^{G12D}$ (Supplementary Fig. 4a,b). This can be explained by the virtually identical residues in the switch I and II regions

of the Ras family members, but the slightly distinct tertiary conformations, compared to those of Ras proteins (Supplementary Fig. 4c,d)[33]. However, RT11 did not cross-react with both active and inactive forms of Rac1[WT], belonging to the Rho family of GTPases[32], which has much different primary and tertiary structures from those of Ras proteins (Supplementary Fig. 4).

**RT11 penetrates into living cells and binds to active Ras**. We next evaluated whether RT11 can reach the cytosol from outside of living cells through a split-GFP (green fluorescent protein) complementation cellular assay[23,24]. In this assay, complemented GFP fluorescence can be observed only when extracellularly-treated RT11-GFP11-SBP2 (RT11 fused with GFP11 fragment and a streptavidin-binding peptide 2 (SBP2) at the C terminus of HC) reaches the cytosol of HeLa-SA-GFP1-10 cells stably expressing a streptavidin (SA)-fused GFP1-10 fragment in the cytosol (Fig. 2a). RT11-GFP11-SBP2 incubated with Ras[WT]-harbouring HeLa-SA-GFP1-10 cells exhibited complemented GFP fluorescence in the cytosol, similar to that observed with TMab4-GFP11-SBP2 (ref. 23), demonstrating that RT11 retains the cytosol-penetrating ability of TMab4 VL. Quantification of the cytosolic access amount[23] of RT11 revealed quantities of ∼14 and 195 nM after 6 h of extracellular treatment of HeLa cells with 0.1 and 1 μM RT11, respectively (Supplementary Table 2).

Furthermore, when incubated with mCherry-KRas[G12V]-transformed mouse fibroblast NIH3T3 and KRas[G12V]-harbouring human colorectal SW480 cells under normal cell culture conditions, internalized RT11, but not TMab4, co-localized with KRas[G12V] at the inner plasma membrane (Fig. 2b), where active KRas[G12V] · GTP is anchored[1,2]. However, in Ras[WT]-harbouring

human colorectal HT29 cells, RT11 predominantly localized throughout the cytosol without enrichment at the plasma membrane (Fig. 2b), similar to TMab4. This could be ascribed to the predominant presence of Ras[WT] in the inactive form in resting cells[34,35]. Immunoprecipitation (IP) using endosome-depleted cell lysates of RT11-treated KRas[G12V]-transformed NIH3T3 cells and KRas[G12V]-harbouring SW480 cells, using RT11 itself, revealed the physical interaction of RT11 with cytosolic KRas mutants after internalization (Fig. 2c). These results demonstrate that RT11, originating from outside the cells, can specifically recognize the GTP-bound active Ras forms at the inner plasma membrane after cellular internalization and cytosolic localization.

**RT11 inhibits the *in vitro* growth of Ras mutant tumour cells**. We next assessed the ability of RT11 to inhibit the proliferation of tumour cells carrying various oncogenic Ras mutants under monolayer culture conditions. In contrast to TMab4, which showed negligible cytotoxicity to cells, RT11 exhibited dose-dependent anti-proliferative activity for diverse oncogenic Ras mutant tumour cells, such as colorectal carcinoma SW480 (KRas[G12V]) and LoVo (KRas[G13D]) cells, pancreatic carcinoma AsPC-1 (KRas[G12D]) and PANC-1 (KRas[G12D]) cells, fibrosarcoma HT1080 (NRas[Q61K]) cells, lung carcinoma H1299 (NRas[Q61K]) cells, leukemic HL60 (NRas[Q61L]) cells and KRas[G12V]-transformed NIH3T3 cells (Fig. 3a and Supplementary Fig. 5a). The anti-proliferative activity of RT11 was much weaker than and comparable to that of the pharmacological inhibitor sorafenib (Raf kinase inhibitor) and LY294002 (PI3K-Akt inhibitor), respectively, when compared at the equivalent molar concentrations (Supplementary Fig. 5b). However, unlike the

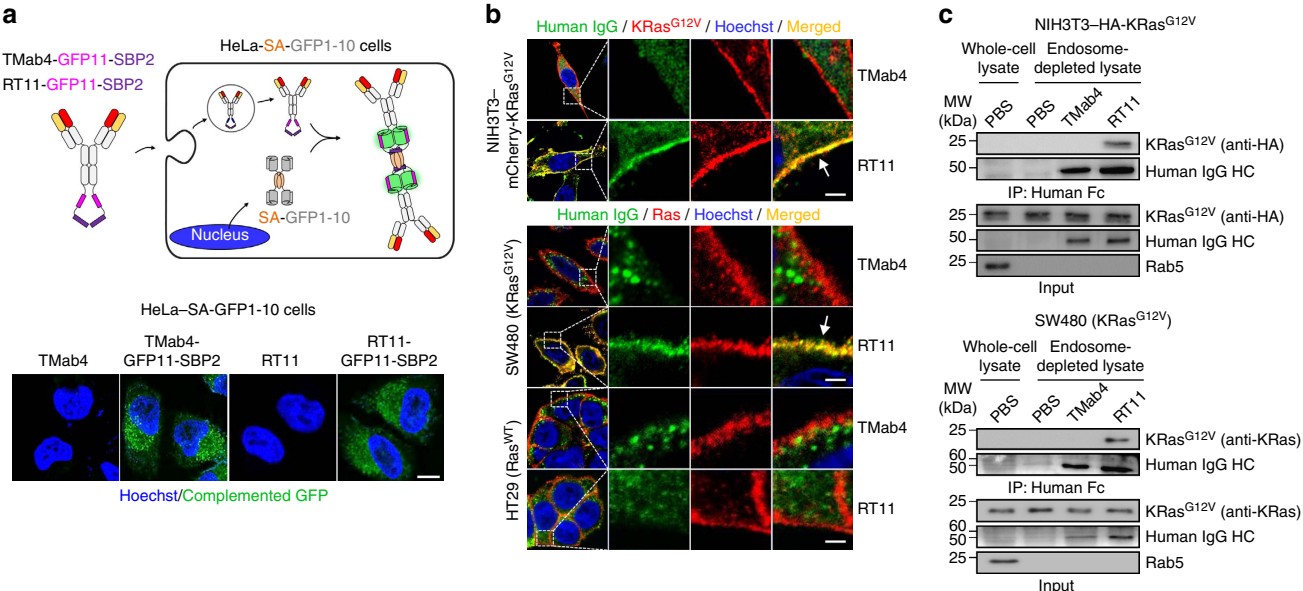

**Figure 2 | RT11 internalizes into the cytosol and binds to active Ras mutant in living cells.** (**a**) Cellular internalization and cytosol localization of GFP11-SBP2-fused RT11 and TMab4 antibodies, as assessed by confocal microscopy measuring complemented GFP signals (green) in HeLa-SA-GFP1-10 cells after treatment with 1 μM of the antibodies for 6 h. Scale bar, 20 μm. (**b**) Cellular internalization and co-localization of RT11 (green), but not TMab4 (green), with the inner plasma membrane-anchored active Ras (red) in mCherry-KRas[G12V]-transformed NIH3T3 and KRas[G12V]-harbouring SW480 cells, analysed by confocal microscopy. The Ras[WT]-harbouring HT29 cells were also analysed as a control. The areas in the white boxes are shown at a higher magnification for better visualization. The arrow indicates the co-localization of RT11 with activated Ras. The cells were treated with 2 μM of antibody for 12 h. Scale bar, 5 μm. In **a,b**, nuclei are counterstained with Hoechst33342 (blue). (**c**) Immunoprecipitation (IP) of KRas mutant with RT11, but not TMab4, from endosome-depleted cell lysates of HA-KRas[G12V]-transformed NIH3T3 and SW480 cells, treated with 2 μM of antibody for 12 h before analysis. The endosome-depleted cell lysates were assessed by the absence of Rab5, an early endosome marker[24]. In **a–c**, images are representative of at least two independent experiments.

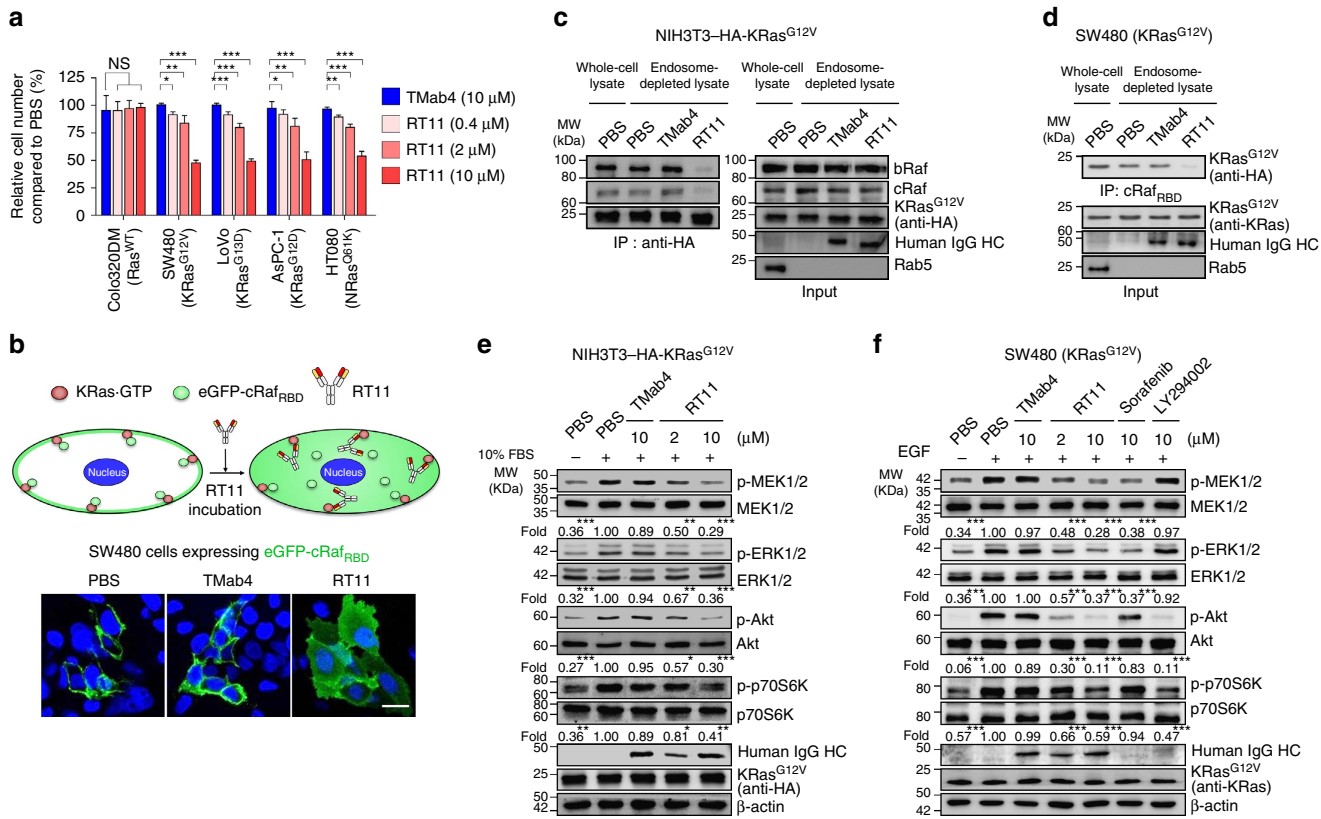

**Figure 3 | RT11 inhibits the growth of oncogenic Ras mutant tumour cells by blocking PPIs between Ras and effector proteins.** (**a**) Cellular proliferation assay, after cells were treated twice at 0 and 72 h with antibody at the indicated concentrations for 6 d. Error bars ± s.d. ($n=3$). $*P<0.05$, $**P<0.01$, $***P<0.001$ versus TMab4-treated cells; NS, not significant. (**b**) Intracellular distribution of eGFP-fused cRaf$_{RBD}$ protein (green) in eGFP-cRaf$_{RBD}$-transformed SW480 cells, treated with antibody (2 μM) for 12 h before microscopic confocal analysis. Scale bar, 20 μm. (**c,d**) IP of endogenous Raf proteins (bRaf and cRaf) with HA-tagged KRas$^{G12V}$ from the endosome-depleted cell lysates of HA-KRas$^{G12V}$-transformed NIH3T3 cells (**c**) and IP of endogenous KRas$^{G12V}$ with cRaf$_{RBD}$ from the endosome-depleted cell lysates of SW480 cells (**d**). The cells were treated with 2 μM of RT11 and TMab4 for 12 h before analysis. (**e,f**) Inhibitory effect of RT11 on the downstream signalling of KRas-effector PPIs in HA-KRas$^{G12V}$-transformed NIH3T3 cells (**e**) and SW480 cells (**f**), analysed by western blotting. The cells were serum-starved for 6 h before treatment with antibody, Raf kinase inhibitor sorafenib, or PI3K-Akt inhibitor LY294002 for 6 h in serum-free growth medium. Cells were washed and then stimulated with 10% FBS (**e**) and EGF (50 ng ml$^{-1}$ in serum free-media) (**f**) for 10 min before cell lysis. The number below the panel indicates relative value of band intensity of phosphorylated proteins compared to that in the PBS-treated control after normalization to the band intensity of respective total protein for each sample. $*P<0.05$, $**P<0.01$, $***P<0.001$ versus PBS-treated control cells. In **b–f**, images are representative of at least two independent experiments.

pharmacological inhibitors, RT11 did not show significant cytotoxicity to Ras$^{WT}$ cells, specifically colorectal Colo320DM and HT29 cells, breast MCF-7 cells and leukemic K562 cells, as well as non-transformed NIH3T3 cells (Fig. 3a and Supplementary Fig. 5a), indicating that direct Ras blocking by RT11 results in minimal toxicity to Ras$^{WT}$ cells probably due to their minimal dependence on the Ras-driven signalling for proliferation[1,2,35]. Oncogenic Ras mutations are known to drive anchorage-independent tumour growth, another important hallmark of cellular transformation[21,36]. On the basis of soft agar colony-formation assays, RT11 treatment resulted in ∼44–64% inhibition of anchorage-independent proliferation for oncogenic KRas mutant cells (SW480, LoVo and PANC-1) and NRas mutant cells (HT1080 and H1229), but not in Ras$^{WT}$ Colo320DM and K562 cells, compared to that after TMab4 treatment (Supplementary Fig. 5c). This result demonstrates that RT11 suppresses the tumorigenic activity of oncogenic Ras mutants. Of note, the effect on tumour growth after blocking oncogenic Ras with RT11 was more evident in anchorage-independent growth conditions (soft agar) than that in monolayer culture conditions (Supplementary Fig. 5a,c), which is line with previous observations with siRNA-mediated knockdown of oncogenic Ras[37,38].

**RT11 blocks active Ras-effector protein interactions**. To elucidate the molecular mechanisms underlying the anti-proliferative activity of RT11, we first examined the ability of RT11 to disrupt the association between oncogenic KRas mutants and effector proteins within the cytosol of cells by analysing the subcellular localization of enhanced GFP-fused cRaf$_{RBD}$ (eGFP-cRaf$_{RBD}$) after transformation into KRas$^{G12V}$-harbouring SW480 cells[10]. Cytosolically expressed eGFP-cRaf$_{RBD}$ was predominantly detected in the inner plasma membrane of cells (Fig. 3b), indicating a physical association between KRas$^{G12V}$ and eGFP-cRaf$_{RBD}$. Noticeably, extracellular treatment of eGFP-cRaf$_{RBD}$-transformed SW480 cells with RT11, but not TMab4, displaced the eGFP-cRaf$_{RBD}$ probe from the inner plasma membrane (location of active KRas) to the cytosol (Fig. 3b). Furthermore, pull-down of hemagglutinin (HA)-tagged KRas$^{G12V}$ with an anti-HA antibody demonstrated reduced co-immunoprecipitation of bRaf and cRaf proteins in RT11-treated HA-KRas$^{G12V}$-transformed NIH3T3 cells (Fig. 3c), and pull-down of active KRas$^{G12V}$ with cRaf$_{RBD}$ resulted in reduced co-immunoprecipitation of KRas$^{G12V}$ in RT11-treated SW480 cells (Fig. 3d), compared to that in TMab4-treated cells. Taken together, the above results demonstrate that RT11 orthosterically

blocks PPIs between active oncogenic KRas mutants and effector proteins after cellular internalization in living cells.

We next investigated the effect of RT11 on downstream signalling mediated by Ras·GTP-effector PPIs such as the Raf-MEK1/2-ERK1/2 and PI3K-Akt-mTOR pathways[1,2]. RT11 exhibited dose-dependent inhibition of downstream signalling mediated by PPIs of Ras-Raf (MEK1/2 and ERK1/2) and Ras-PI3K (Akt and p70S6K) in serum-stimulated HA-KRas$^{G12V}$-transformed NIH3T3 cells (Fig. 3e) and epidermal growth factor (EGF)-stimulated SW480 cells (Fig. 3f). The Raf inhibitor sorafenib and PI3K-Akt inhibitor LY294002 attenuated only targeted signalling in SW480 cells (Fig. 3f). These results demonstrate that competitive blocking of active Ras-effector PPIs by RT11 results in the attenuation of downstream signalling, which is required for cell growth and survival.

We examined whether RT11 affects the intrinsic GTPase activity of GTP-loaded KRas$^{WT}$ and KRas$^{G12D}$ in vitro. RT11 did not significantly affect the intrinsic GTP hydrolysis rate of both KRas$^{WT}$ and KRas$^{G12D}$, similar to TMab4 (Supplementary Fig. 6), indicating that the inhibitory activity of RT11 against active Ras signalling is not through an acceleration in GTP hydrolysis of active Ras.

**Generation of tumour-associated integrin targeting RT11-i.** Since RT11 internalizes through HSPG, which is ubiquitously expressed in epithelial cells[22], we hypothesized that this antibody might lack tissue specificity when administered systemically. To confer tumour tissue specificity to RT11, the RGD10 cyclic peptide (13 residues: DGARYCRGDCFDG), which binds specifically to tumour-associated integrins such as αvβ3 and αvβ5 (ref. 39), was genetically fused to the N-terminus of the LC of RT11 via a (G$_4$S)$_2$ linker, generating RT11-i (Fig. 4a). As a control, TMab4-i was also generated in the same manner. Integrin αvβ3 and/or αvβ5 are overexpressed on the surface of many types of tumour cells and tumour-associated blood vessels[40]. RT11-i and TMab4-i, purified in well-assembled IgG forms without non-native soluble oligomers (Supplementary Fig. 7a,b), selectively bound to integrin αvβ3-transformed K562 cells, but not to the WT K562 cells, as well as to integrin αvβ5-expressing tumour cells (Fig. 4b,c). Furthermore, RT11-i retained its selective binding of only active Ras forms (Supplementary Fig. 7c), its cytosol-penetrating ability (Supplementary Fig. 7d and Supplementary Table 2), and its binding to inner plasma membrane-anchored active KRas mutants after cellular internalization (Fig. 4d,e). Even after 48 h of incubation at 50 °C, both RT11 and RT11-i maintained selective binding to active KRas$^{G12D}$ form without the formation of soluble oligomers (Supplementary Fig. 8), indicating that RT11-i possesses comparable thermal stability to that of RT11. Thus, the fusion of the RGD10 peptide to the N terminus of the LC of RT11 did not compromise the biochemical and biophysical properties of RT11.

RT11-i exhibited significantly improved anti-proliferative activity against oncogenic Ras mutant tumour cells grown in monolayer culture conditions, but not against Ras$^{WT}$ Colo320DM cells, when compared with that of TMab4-i (Supplementary Fig. 7e). However, the Ras-specific blocking effect was modest because TMab4-i itself also exerted anti-proliferative activity due to RGD10-mediated integrin blocking of anchorage-dependent growth[39,40]. When assessed in anchorage-independent growth conditions on soft agar, RT11-i resulted in ∼40–67% suppression of colony formation in oncogenic Ras mutant SW480 (KRas$^{G12V}$), LoVo (KRas$^{G13D}$), AsPC-1 (KRas$^{G12D}$), PANC-1 (KRas$^{G12D}$), HT1080 (NRas$^{Q61K}$) and H1299 (NRas$^{Q61K}$) cells, but not in Ras$^{WT}$ Colo320DM cells,

when compared to TMab4-i (Fig. 4f and Supplementary Fig. 7f). These data demonstrate that RT11-i retains the anti-proliferative activity of RT11 by specifically blocking oncogenic Ras signalling in tumour cells.

**Pharmacokinetics and biodistribution of RT11-i.** The pharmacokinetic profiles of RT11-i and TMab4-i were examined in BALB/c athymic nude mice following a single intravenous injection of 20 mg kg$^{-1}$. The plasma concentrations of the antibodies declined in a biphasic manner (Fig. 5a), which is a typical profile for systemically administered antibodies[41]. The terminal serum half-life (T$_{1/2}$β) was similar for each antibody, and was specifically 105.8 ± 2.3 h and 100.1 ± 3.1 h for RT11-i and TMab4-i, respectively, resembling that of other IgG-format antibodies in mice[41].

We next determined tissue distribution by intravenous dosing of DyLight755-labelled antibodies into nude mice bearing integrin αvβ5-expressing SW480 xenograft tumours. RT11 without RGD10 fusion did not exhibit any increased distribution in the tumours, compared to that in normal tissues, during 24 h circulation (Supplementary Fig. 9a,b). In contrast, RGD10-fused RT11-i and TMab4-i displayed preferential accumulation in the tumours with a peak at 24 h post injection, when compared to that in the normal tissues (Fig. 5b and Supplementary Fig. 9c), demonstrating the in vivo tumour targeting ability of RT11-i and TMab4-i.

**RT11-i inhibits the in vivo growth of Ras mutant tumours.** We next assessed the in vivo anti-tumour efficacy of RT11-i through intravenous injection into mice harbouring pre-established oncogenic Ras mutant tumour xenografts (SW480 (KRas$^{G12V}$), LoVo (KRas$^{G13D}$), and HT1080 (NRas$^{Q61K}$)), or Ras$^{WT}$ tumours (Colo320DM). Compared to the PBS-treated vehicle control, TMab4-i slightly retarded the growth of SW480 and HT1080 tumours, which was accompanied by reduced phosphorylation of Akt (Fig. 6a,b and Supplementary Figs 10 and 11); this could be attributed to the anti-tumour activity of the RGD10 moiety, through blocking integrin αvβ3- and/or αvβ5-mediated tumour angiogenesis and growth[39,40]. Nonetheless, the growth of LoVo tumours was not affected by TMab4-i. Importantly, RT11-i markedly inhibited the growth of the three oncogenic KRas mutant tumour xenografts, showing ∼46–70% more tumour-growth inhibition (TGI) (Fig. 6a) and ∼42–63% greater reduction in tumour weight, compared to those after treatment with TMab4-i (Supplementary Fig. 10b). However, in case of Ras$^{WT}$ Colo320DM tumours, no significant difference was observed in anti-tumour efficacy between RT11-i and TMab4-i. During the antibody treatments the mice did not exhibit any significant body weight loss (Supplementary Fig. 10c). Thus, the additional anti-tumour activity of RT11-i for Ras mutant tumours could be ascribed to the blocking activity of oncogenic Ras signalling. The in vivo sensitivity of the tumours to RT11-i was in the following order: HT1080 (∼70% TGI) > SW480 (∼56% TGI) > LoVo (∼46% TGI), which roughly correlated with in vitro colony formation assay results (Fig. 4f), indicating their different dependency on oncogenic Ras signalling.

The examination of excised tumour tissues following treatment showed co-localization of RT11-i with oncogenic Ras mutants at the inner plasma membrane of tumour cells, whereas TMab4-i was observed only in the cytoplasmic regions without co-localization with Ras mutants (Fig. 6c and Supplementary Fig. 11a). This result indicates that RT11-i is bioavailable in vivo inside the cytosol of tumour cells after systemic intravenous administration. Compared to those in vehicle- and TMab4-i-treated tumour tissues, RT11-i-treated tumour tissues

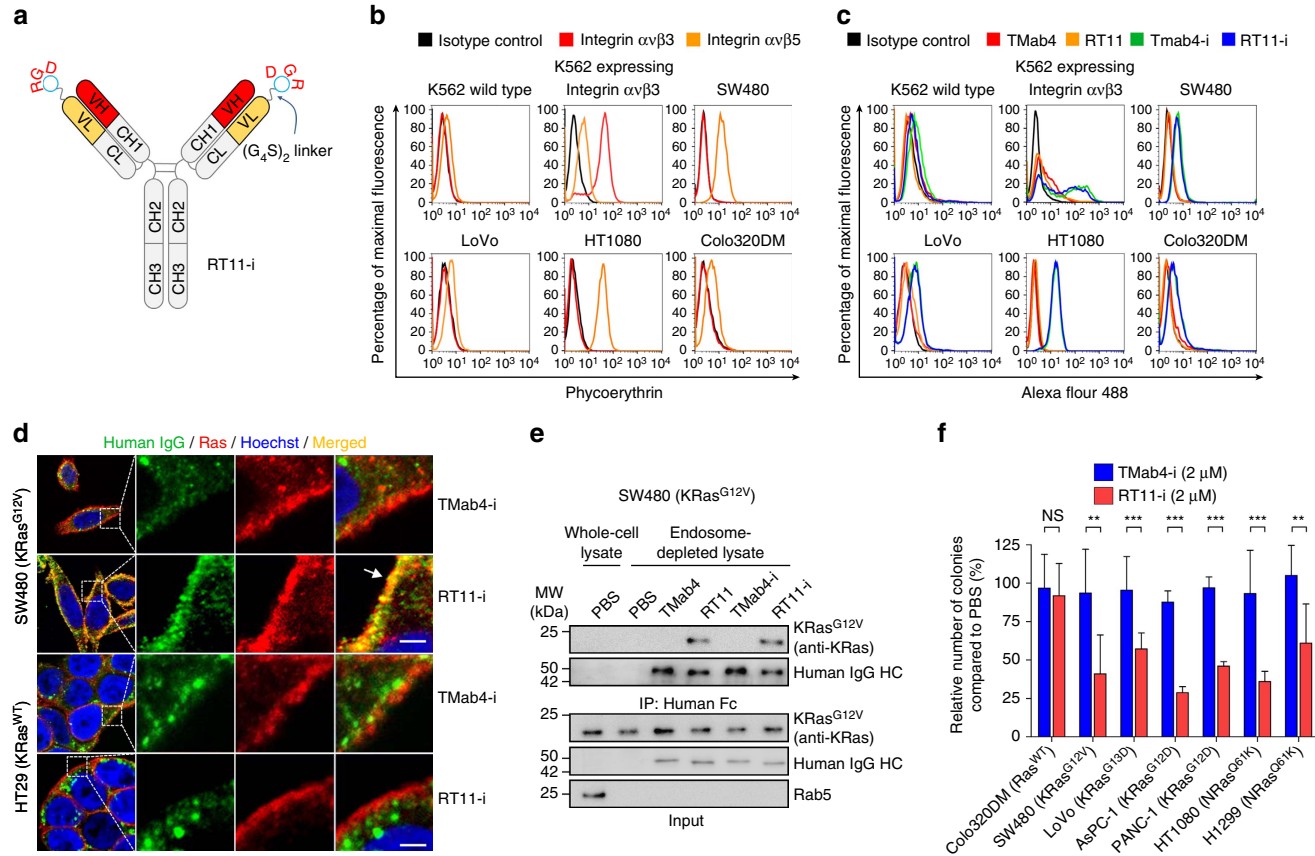

**Figure 4 | Generation and characterization of RGD10 peptide-fused RT11-i iMab. (a)** Generation of integrin $\alpha v\beta 3/\alpha v\beta 5$-targeting RT11-i by genetic fusion of RGD10 peptide, using a $(G_4S)_2$ linker, to the N-terminus of the LC of RT11. **(b,c)** RT11-i and TMab4-i bind to cell surface-expressed integrin $\alpha v\beta 3$ and $\alpha v\beta 5$. In **b**, flow cytometric analysis of the cell surface expression levels of integrin $\alpha v\beta 3$ and $\alpha v\beta 5$ on WT K562, integrin $\alpha v\beta 3$-transformed K562, and human tumour cells, analysed by PE-conjugated anti-human integrin $\alpha v\beta 3$ and $\alpha v\beta 5$ antibodies. In **c**, flow cytometric analysis of cell surface binding levels of the indicated antibodies, co-incubated at 100 nM with 300 IU ml$^{-1}$ heparin for 1 h at 4 °C with the indicated cells before analysis. **(d)** Cellular internalization and co-localization of RT11-i, but not TMab4-i, with the inner plasma membrane-anchored active Ras · GTP in KRas$^{G12V}$-harbouring SW480 cells. The Ras$^{WT}$-harbouring HT29 cells were also analysed as a control. The areas in the white boxes are shown at increased magnification for better visualization. The arrow indicates the co-localization of RT11-i with activated Ras. Nuclei were counterstained with Hoechst 33342 (blue). Scale bar, 5 µm. **(e)** IP of endogenous KRas$^{G12V}$ with RT11 or RT11-i, but not TMab4 and TMab4-i, from endosome-depleted cell lysates of SW480 cells. Images are representative of two independent experiments. In **d,e**, the cells were treated with 1 µM of antibodies for 12 h before analysis. **(f)** Inhibition of tumour cell soft agar colony formation by RT11-i compared to that with TMab4-i. Following treatment of cells with PBS, TMab4-i (2 µM), or RT11-i (2 µM) every 72 h for 2–3 weeks, the number of colonies (diameter > 200 µm) was counted by BCIP/NBT staining, as shown in the pictures of representative soft agar plates (Supplementary Fig. 7f). The results are presented as percentages compared to the PBS-treated control. Error bars represent the mean ± s.d. (n = 3). **P < 0.01, ***P < 0.001; NS, not significant.

showed significantly reduced levels of phosphorylated MEK1/2, ERK1/2 and Akt (Fig. 6b and Supplementary Fig. 11b,c) and decreased staining for the cell proliferation marker Ki-67, as well as increased apoptosis as measured by TUNEL-staining, only for oncogenic Ras mutant tumour cells (Supplementary Fig. 11d,e). Taken together, the above results identified the *in vivo* anti-tumour mechanism of RT11-i against oncogenic Ras mutant-driven tumour growth.

**RT11-i overcomes KRas mutation-driven cetuximab resistance.** Oncogenic KRas mutations have been associated with anti-EGFR antibody resistance in adenocarcinoma such as colon cancer[42,43]. We examined whether combined treatment with RT11-i and the anti-EGFR antibody cetuximab (Erbitux) could overcome cetuximab resistance in colorectal LoVo tumours harbouring the oncogenic KRas$^{G13D}$ mutation, using a xenograft mouse model. Biweekly injections of cetuximab alone at a high dose (50 mg kg$^{-1}$) or that in combination with TMab4-i only slightly impaired tumour growth compared to that in vehicle-treated

control mice (Fig. 7a–c), confirming the poor responsiveness of LoVo xenografts to cetuximab treatment[44]. However, combined treatment of cetuximab plus RT11-i dramatically retarded tumour growth and increased mouse survival, by ∼16 d, compared to those in mice treated with cetuximab alone or cetuximab plus TMab4-i. Analysis of excised tumour tissues following treatment revealed that combined treatment dramatically reduced the phosphorylation of ERK1/2, Akt, and STAT3, compared to that with cetuximab alone treatment (Fig. 7d). These results suggest that RT11-i inhibits oncogenic KRas mutant-driven signalling, thereby overcoming cetuximab resistance in KRas mutant colorectal tumours.

## Discussion

Many disease-associated intracellular proteins have been considered undruggable using conventional antibody regimens. In this study, we demonstrated that intracellular oncogenic Ras mutants could be directly targeted, to block their function, from outside of living cells using RT11/RT11-i iMab antibodies.

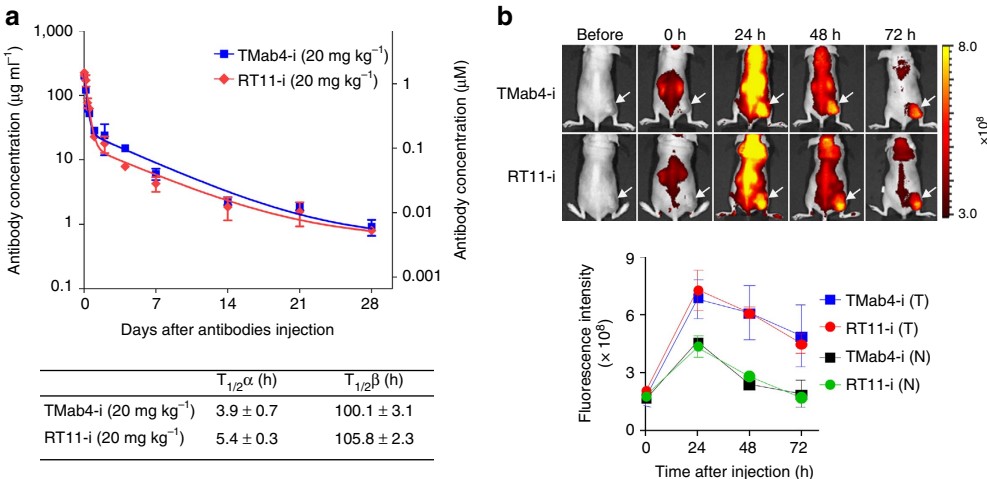

**Figure 5 | Pharmacokinetics and biodistribution of RT11-i and TMab4-i antibodies.** (**a**) Pharmacokinetic profiles of RT11-i and TMab4-i in non-tumour bearing mice. Serum concentrations of TMab-i and RT11-i were determined by ELISA in female BALB/c nude mice following a single intravenous injection of 20 mg kg$^{-1}$ in a total volume of 200 μl. Error bars represent the mean ± s.d. ($n = 3$ per time point). The solid lines represent the fit of a two-compartment pharmacokinetic model to the data to estimate the initial rapid clearance phase ($T_{1/2}\alpha$) and later terminal serum clearance phase ($T_{1/2}\beta$). The inset table shows the pharmacokinetic parameters. (**b**) Tumour-targeting ability of RT11-i and TMab4-i, evaluated by intravenously injecting Dylight755-labelled antibodies (20 μg per mouse) into SW480 xenograft tumour-bearing mice, followed by *in vivo* fluorescence imaging. Representative images are shown, which were acquired at the indicated times post-injection. Fluorescence intensities in the tumour tissue (T), as indicated by arrows, and normal tissues (N) were quantified by radiant efficiency (photons s$^{-1}$ cm$^{-2}$ steradian$^{-1}$ μW$^{-1}$ cm$^{-2}$) using Living Image software. Error bars, ± s.d. ($n = 5$ per group).

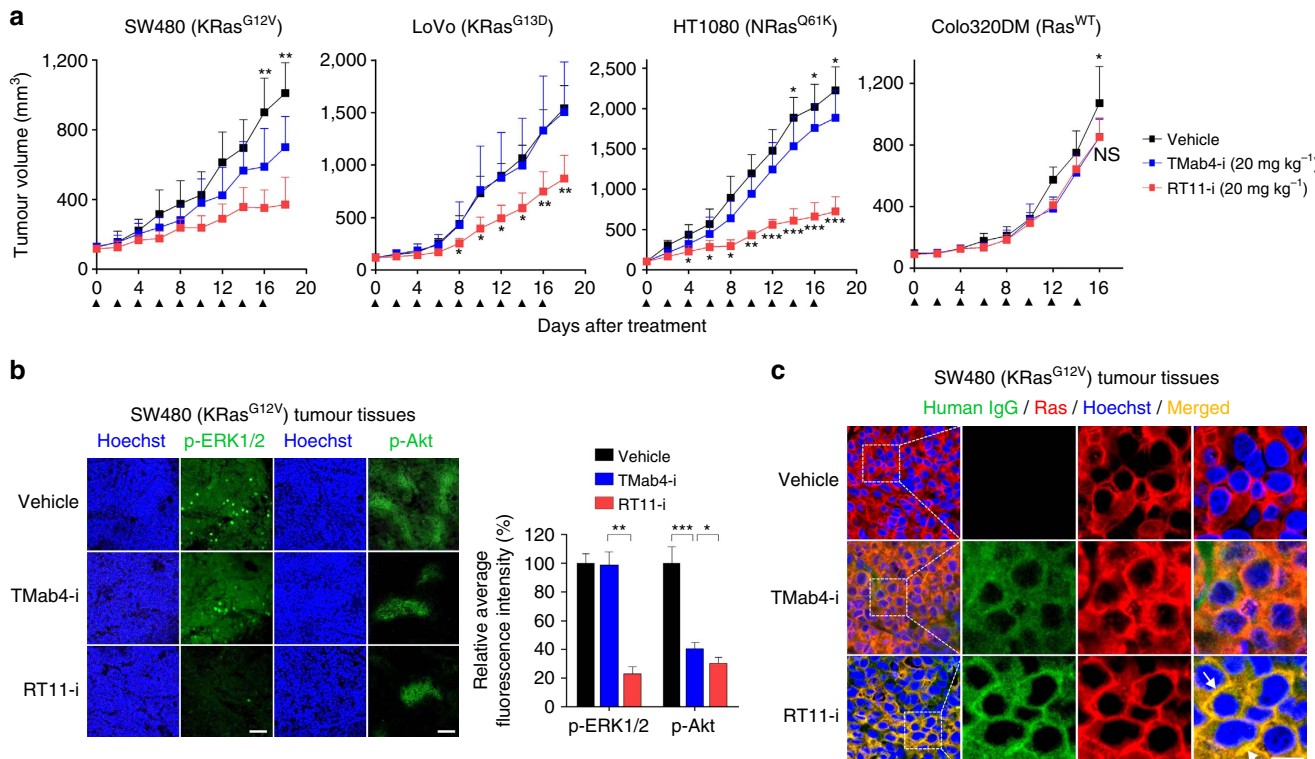

**Figure 6 | RT11-i suppresses the *in vivo* growth of oncogenic Ras mutant tumour xenografts in mice.** (**a**) *In vivo* anti-tumour efficacy of RT11-i compared to that of vehicle and TMab4-i controls, analysed by measuring the tumour volume during treatment of female BALB/c nude mice harbouring the indicated tumour xenografts. Antibodies were intravenously dosed at 20 mg kg$^{-1}$ every 2 d (indicated by the arrows). Error bars, ± s.d. ($n = 8$ per group). (**b,c**) Immunohistochemical images showing the levels of p-ERK1/2 and p-Akt (**b**) or cellular penetration and co-localization of antibodies with the active Ras form (**c**) in SW480 tumour tissues excised from mice following treatment described in **a**. Images are representative of three independent experiments. Nuclei were counterstained with Hoechst33342 (blue). Scale bar, 100 μm in **b** or 10 μm in **c**. In **b**, the right panel shows the percent relative fluorescence intensity compared to that of vehicle-treated control. Error bars, ± s.d. of five random fields for each tumour (two tumours per group). In **c**, the areas in the white boxes are shown at a higher magnification for better visualization. The arrows indicate the co-localization of RT11-i with activated Ras. In **a,b**, statistical analysis was performed using a one-way analysis of variance followed by the Newman–Keuls post-test. *$P < 0.05$, **$P < 0.01$, ***$P < 0.001$ versus TMab4-i; NS, not significant.

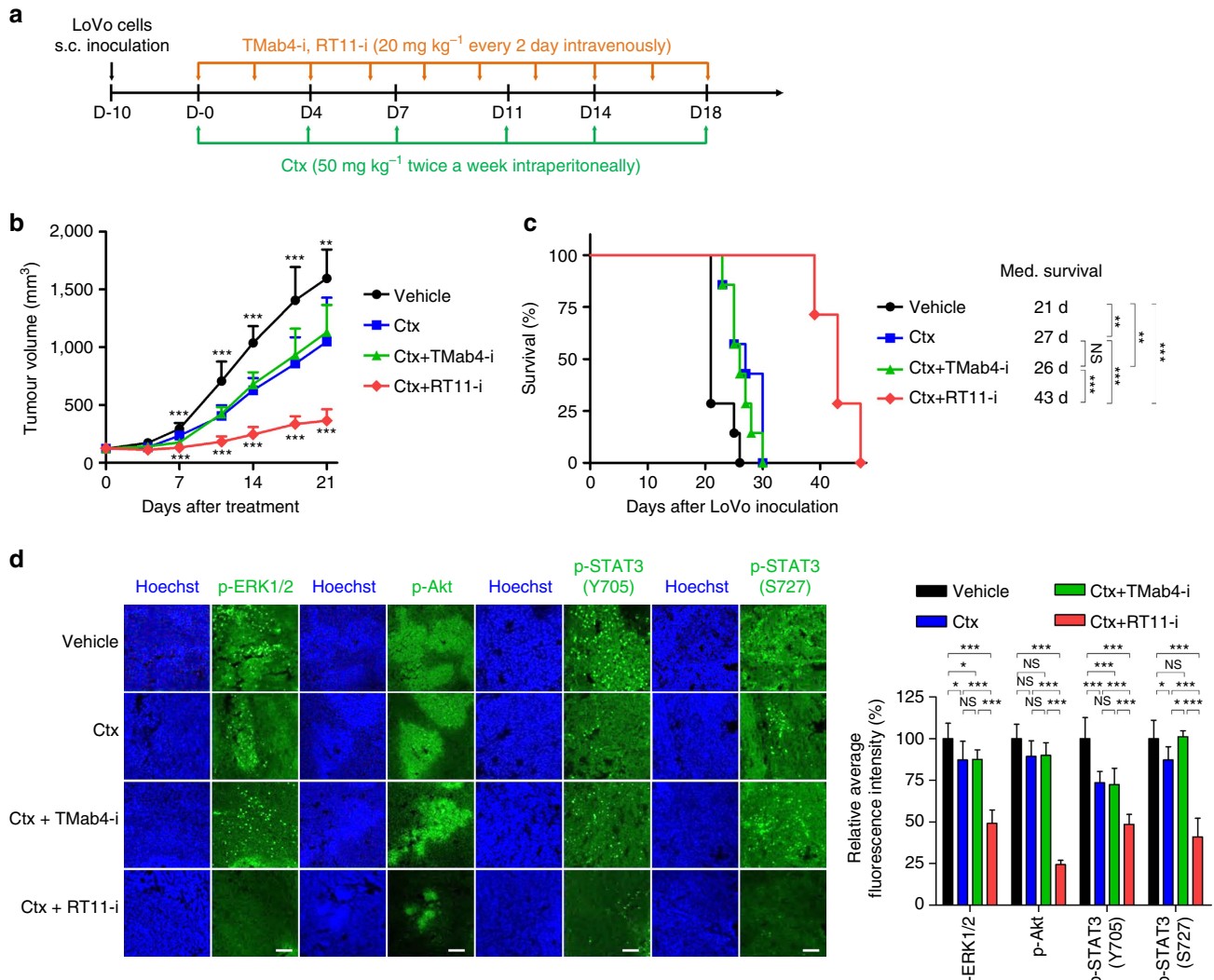

**Figure 7 | Co-treatment of RT11-i overcomes cetuximab resistance in KRas mutant colorectal LoVo tumours in mice.** (**a**) Outline of the RT11-i, TMab4-i, and/or cetuximab treatment regimen. (**b**) Tumour growth was analysed by measuring the tumour volume during treatment with vehicle, cetuximab (Ctx), Ctx plus TMab4-i or Ctx plus RT11-i in Lovo xenograft mice. Error bars, ± s.d. (*n* = 8 per group). Statistical analysis was performed using a one-way analysis of variance followed by the Newman-Keuls post-test. ***P < 0.001 versus Ctx alone. (**c**) Kaplan–Meier survival curves with median survival time listed for LoVo xenograft mice for the vehicle, Ctx, Ctx + TMab4-i and Ctx + RT11-i treatment groups (*n* = 8 per group). **P < 0.01, ***P < 0.001 by Gehan-Breslow-Wilcoxon test for significance. Animals were killed when tumours reached ~1,000 mm³ in size. (**d**) Immunohistochemical analysis showing the levels of p-ERK1/2, p-Akt, p-STAT3 (Y705) and p-STAT3 (S727) in LoVo tumour tissues excised from mice following the treatment described in **b**. Nuclei were counterstained with Hoechst 33342 (blue). Scale bar, 100 μm. Images are representative of at least two independent experiments. The right panel shows the percent relative fluorescence intensity compared to that of the vehicle-treated control group. Error bars, ± s.d. of five random fields for each tumour (two tumours per group). Statistical analysis was performed using a one-way analysis of variance followed by the Newman–Keuls post-test. *P < 0.05, **P < 0.01, ***P < 0.001; NS, not significant.

Systemic administration of RT11-i resulted in significant growth inhibition and apoptosis in oncogenic Ras mutant-harbouring tumour xenografts in mice, without significant toxicity to Ras^WT-carrying tumours. This implies its therapeutic potential as an anti-cancer antibody specific for oncogenic Ras mutant tumours. Importantly, our study illustrates the development of a new class of antibody, iMab, the action of which can be dissected into four sequential steps: (1) tumour homing by targeting tumour-associated cell surface-expressed proteins after systemic administration, (2) internalization by receptor-mediated endocytosis into targeted tumour cells, (3) cytosolic localization owing to their ability to escape from early endosomes, and (4) specific binding to a targeted cytosolic protein to block its function.

The RT11 iMab was generated by integrating the Ras · GTP-specific binding VH domain with the cytosol-penetrating VL fragment to form a single IgG format antibody. Competition screening with KRas^G12D · GppNHp in the presence of excess amounts of KRas^G12D · GDP as a competitor resulted in successful isolation of VH-dependent active Ras form–specific RT11, which selectively binds to active forms of both WT and oncogenic mutants of KRas, NRas, and HRas with similar affinities, but not to GDP-bound inactive forms (Fig. 1 and Supplementary Table 1). Importantly, the binding sites of RT11 were mapped to the switch I and switch II regions of the GTP-bound active Ras form, which are conformationally distinct from the GDP-bound inactive forms, namely the commonly shared PPI interfaces of oncogenic Ras mutants with the effector proteins[1,2,6]. This epitope identification explains how RT11/RT11-i competitively blocks the physical interactions between activated Ras and its effector proteins, leading to inhibition of

downstream signalling mediated by Ras-effector PPIs, such as Raf-MEK1/2-ERK1/2 and PI3K-Akt pathways (Fig. 3e,f and Supplementary Fig. 11). These results suggest that blocking Ras·GTP–effector protein PPIs is the molecular mechanism of action for RT11/RT11-i iMabs. Because the four Ras proteins (KRas4A, KRas4B, HRas and NRas) differ markedly in their C-terminal hypervariable regions, but share almost identical PPI interfaces in the switch regions[1,13], RT11/RT11-i is anticipated to exhibit broad specificity to the oncogenic Ras mutants. This was shown using KRas, NRas, and HRas mutants with predominant oncogenic mutations at G12, G13 or Q61 residues (Fig. 1c and Supplementary Fig. 7c). This reasoning was further supported by the anti-proliferative effect of RT11/RT11-i on a panel of tumour cells, including colorectal, pancreatic, and lung carcinoma, fibrosarcoma, and leukemia, harbouring diverse oncogenic Ras mutations (Figs 3a and 4f and Supplementary Figs 5 and 7e,f). The binding characteristics of RT11/RT11-i will ultimately broaden its therapeutic applications through the comprehensive targeting of various oncogenic Ras mutants.

For systemic in vivo applications, the RGD10 peptide-fused RT11-i was generated for tumour tissue homing ability, via targeting tumour-associated integrin αvβ3 and/or αvβ5 (refs 39,40). RT11-i exhibited selective binding to the cell surface-expressed integrin αvβ3/αvβ5 while retaining the binding specificity of RT11 to active Ras after cellular internalization. Importantly, when intravenously administered, RT11-i demonstrated reasonable serum half-life and preferential accumulation in tumour tissue (Fig. 5). RT11-i might exert some anti-tumour activity due to the RGD10-mediated integrin blocking activity[39,40], particularly for integrin αvβ3/αvβ5-overexpressing tumour cells, based on the effects of TMab4-i on SW480 and HT1080 xenograft tumours (Fig. 6a,b). However, RT11-i as a single agent exhibited measurable anti-tumour activity via the oncogenic Ras-specific blocking mechanism, showing a ∼46–70% more TGI, when compared to treatment with TMab4-i, in mice harbouring oncogenic KRas mutant SW480 and LoVo tumours as well as NRas mutant HT1080 tumours, but not in those harbouring Ras^WT tumours (Fig. 6), without systemic toxicity (Supplementary Fig. 10c). Furthermore, in xenograft mouse models, RT11-i sensitized cetuximab-resistant, KRas^G13D mutant colorectal LoVo tumours to cetuximab by inhibiting the bypass oncogenic KRas mutant signalling (Fig. 7). This suggests that the combination of RT11-i and an anti-EGFR antibody might be an effective clinical strategy for patients with advanced colon cancer harbouring oncogenic KRas mutations. Since the RT11-i iMab has many desirable features of conventional IgG antibodies, including the capacity for large-scale manufacturing, systemic administration and long serum half life, it has great potential to be developed as a first-in-class anticancer antibody.

Nonetheless, the in vivo anti-tumour activity of RT11-i is not very potent as a single agent, and requires frequent dosing at relatively high concentrations for significant suppression of tumour growth (Fig. 6). Thus RT11-i should be further engineered to improve potency for practical clinical studies. For this purpose, the endosome escape efficiency (currently ∼4.3%) of RT11-i after receptor-mediated endocytosis requires improvement to achieve sufficient cytosolic localization to address relatively high cellular concentrations (0.4–1.6 μM) of Ras[45]. Such RT11-i derivative can be generated by incorporating the recently engineered cytosol-penetrating TMab4 VL variant with ∼threefold improved endosomal escape efficiency (∼13%)[24]. Furthermore, affinity maturation of RT11-i against the active form of oncogenic Ras mutants (currently 4–17 nM) will be necessary to increase the efficiency of blocking Ras·GTP–effector PPIs.

In conclusion, our results demonstrate the feasibility of developing antibody therapeutics that directly target cytosolic proteins involved in disease-associated PPIs, such as oncogenic Ras mutants, by systemic administration, similar to conventional therapeutic antibody regimens. Our work also provides an innovative antibody platform technology, iMab, which can extend the antibody target space to cytosolic proteins from the current limits of only extracellular proteins.

## Methods

**Antibodies.** Antibodies used as reagents in this study are listed in Supplementary Table 3.

**Protein expression and purification.** Plasmids including pGEX-3X-HRas^G12V, encoding human HRas^G12V, and pEF-BOS-HA-KRas^G12V, encoding human KRas^G12V, were kindly provided by Professor T.H. Rabbitts (St James's University Hospital, UK)[20] and Prof T. Kataoka (Kobe University, Japan)[9], respectively. Human NRas (clone ID hMU009336), RRas (clone ID hMU001042), Rac1 (clone ID KU000510), RasGAP (RASA1) (clone ID hMU002576) genes were purchased from Korea human gene bank of Medical Genomics Research Center (KRIBB, Korea). Human MRas (residues 1–208, Genebank accession code: NM_012219.4), cRaf_RBD (residues 1–149, GeneBank accession code: NM_002880.2), and RalGDS_RBD (residues 788-885, GeneBank accession code: NM_006266) genes were prepared by DNA synthesis (Bioneer Inc., Korea). For the bacterial expression of KRas^WT (residues 1–188), HRas^WT (residues 1–189), and NRas^WT (residues 1–189) and their oncogenic mutants, cRaf_RBD (residues 1–149), and RalGDS_RBD (residues 788-885), the respective gene fragment was cloned in frame into pGEX-3X vector (GE Healthcare) to be expressed in the N-terminal GST-fused form. The GST-fused proteins were expressed in E. coli BL21 (DE3)plysE cells and purified using Glutathione-Sepharose resin (GE Healthcare) according to the manufacturer's. The purified GST-fused Ras proteins were used in ELISA, surface plasmon resonance and/or GTPase assays. For the bacterial expression of KRas^G12D (residues 1–169)[8], its alanine-scanning mutants, MRas (residues 1–208), RRas (residues 1–218), and Rac1 (residues 1–177), the respective gene fragment was cloned into pET23 vector (Novagen) to be expressed in the N-terminal 6 × His-fused form. The His-fused proteins were expressed in E. coli BL21 (DE3)plysE cells and purified using Ni-NTA resin (Clontech) according to the manufacturer's instructions. The purified His-fused KRas^G12D (residues 1–169) was used for antibody library screening by MACS and FACS, competition ELISA with cRaf_RBD and RalGDS_RBD for KRas binding, and/or antibody epitope mapping studies. The purified His-fused MRas, RRas, and Rac1 were used for ELISA. For mammalian expression of RasGAP, a DNA encoding the catalytic domain of RasGAP (residues 714–1047) was cloned in frame into pSecTag2A vector (Invitrogen)[22] to be expressed in the N-terminal 6 × His-fused form. The plasmid was transiently transfected into HEK293F cell culture in Freestyle 293F (Invitrogen) following the standard protocol[22,24]. RasGAP was purified using Ni-NTA resin (Clontech). Protein concentrations were determined using a Bicinchoninic Acid (BCA) kit (Pierce, 23225) and by measuring the absorbance at 280 nm.

**Preparation of GppNHp- or GDP-loaded Ras proteins.** Ras protein was loaded with the non-hydrolysable GTP analogue GppNHp (Sigma-Aldrich, G0635) or GDP (Millipore, 20–177) following the previously reported protocol[30,46]. For GppNHp loading, the purified Ras protein was incubated with a 20-fold molar excess of GppNHp in exchange buffer (40 mM Tris-HCl, pH 7.5, 200 mM (NH4)2SO4, 10 μM ZnCl2, 5 mM DTT) containing alkaline phosphatase beads (2 units per mg of Ras protein) for 1 h at 25 °C with gentle agitation. The reaction mixture was then centrifuged to pellet the alkaline phosphatase beads and the supernatant was taken. For GDP loading, the purified Ras protein was incubated with a 20-fold molar excess of GDP in Ras storage buffer (50 mM Tris-HCl, pH 8.0, 1 mM DTT, 2 mM MgCl2) containing 20 mM EDTA for 30 min at 30 °C and further incubated for 30 min at 4 °C after adding 60 mM MgCl2 to the buffer. To remove the excess unbound GppNHp or GDP from the reaction mixture, the sample was applied to a PD10 Sephadex G25 column (GE Healthcare) equilibrated with Ras storage buffer according to the manufacturer's instructions and elution fraction was collected. For the antibody screening probe, KRas^G12D·GppNHp was further biotinylated using EZ-Link Sulfo-NHS-Biotin (Thermo Scientific, #21217) in accordance with the manufacturer's instructions[47].

**Screening of yeast antibody library against KRas^G12D·GppNHp.** Yeast strains and media composition have been previously described[25,48,49]. The synthetic human HC library displayed in the format of VH library-CH1 on the surface of Saccharomyces cerevisiae JAR200 haploid cells with mating type a (MATa) was used[25,49]. For HC haploid library screening against KRas^G12D·GppNHp, one round of MACS and then one round of FACS were sequentially performed using 1 and 0.5 μM of biotinylated KRas^G12D·GppNHp in the presence of 10-fold excess molar concentrations of non-biotinylated KRas^G12D·GDP as a competitor[25,47,48].

After the first round of FACS, the enriched HC yeast haploid library cells were mated with the other mating type of *S. cerevisiae* YVH10 cells (MATα) that secreted a fixed LC with the VL-CL (LC constant domain) of TMab4 to generate an enriched Fab library on the diploid yeast cells, as illustrated in Fig. 1b. The mated diploid Fab library was screened by FACS using conditions that were more stringent: 0.1 μM KRas$^{G12D}$·GppNHp and 5 μM competitor (1:50 competition) in round 2, 0.1 μM KRas$^{G12D}$·GppNHp and 10 μM competitor (1:100 competition) in round 3. The binding level of biotinylated KRasG12D·GppNHp was observed by Streptavidin-conjugated R-phycoerythrin (SA-PE) (Thermo Scientific, S-21388, dilution 1: 600). Typically the top 0.5–1% of target-binding cells was sorted. The final sorted yeast cells were plated on selective medium and individual clones were isolated. The *VH* gene from selected diploid yeast cells was amplified by colony PCR for sequencing[25,49].

**Affinity maturation of RT4.** Three kinds of HC libraries were generated focusing on VH-CDRs of RT4 by performing serial overlapping PCRs[47], as illustrated in Supplementary Fig. 2. Since VH-CDR3 tends to locate near to the center of the antigen-antibody interface[50], three distinct HC libraries, dubbed library 6, 7 and 9, with the length variation of 6, 7, and 9 residues of VH-CDR3, respectively, were designed mutating each residue with a degenerate codon NNK, which encodes all 20 amino acid with a reduced stop codon frequency[48]. In addition, in all three HC libraries, the exposed residues of VH-CDR1 (residues 31–33), VH-CDR2 (residues 50, 52–56) were commonly randomized, while maintaining the original amino acids of RT4 at a level of ∼50% at each residue, using designed oligomers[47,51]. For the yeast surface-displayed HC library, each amplified VH gene library (12 μg) and linearized yeast surface display vector pYDS-H (4 μg) were co-transformed four times into *S. cerevisiae* JAR200 strain by a homologous recombination technique using a BioRad Gene Pulser II model #1652108 (refs 25,48,49). The diversities of library 6, 7, and 9, determined by plating serial 10-fold dilutions of the transformed cells onto selective agar plates, were ∼4.6 × 10$^8$, 2.0 × 10$^8$ and 2.4 × 10$^8$, respectively. The library screening was conducted as described above under more stringent competition screening conditions.

**Construction of antibody expression plasmids.** The isolated *VH* gene was subcloned, in-frame without additional amino acids, into *Not*1/*Apa*I sites of pcDNA 3.4-TMab4-HC carrying the human IgG1 constant domain sequence (CH1-hinge-CH2-CH3)[22] for HC expression. For LC expression, the pcDNA 3.4-TMab4-LC plasmid encoding the TMab4 VL and Cκ constant domain sequences (residues 108–214) was used[22]. For LC expression of RGD10-fused TMab4-i and RT11-i antibodies, the synthesized DNA encoding the RGD10 peptide (DGARYCRGDCFDG)[39] and (G₄S)₂ linker was subcloned, in frame without additional amino acids, at the N-terminus of TMab4 VL using the *Not*I/*Bsi*WI site of pcDNA 3.4-TMab4-LC, generating pcDNA 3.4-TMab4-iLC. All the constructs were confirmed by sequencing (Macrogen, Inc.).

**Expression and purification of antibodies.** The plasmids encoding HC and LC were transiently co-transfected in pairs, at equivalent molar ratios, into HEK293F cell cultures in Freestyle 293F media (Invitrogen) following the standard protocol[22,52]. Culture supernatants were collected after 5–7 d by centrifugation and filtration (0.22 μm, Polyethersulfone, Corning, CL S43118). Antibodies were purified from the culture supernatants using a protein-A agarose chromatography column (GE Healthcare) and were extensively dialysed to achieve a final buffer composition of PBS (pH 6.5). The anti-EGFR mAb cetuximab (Erbitux) was expressed and purified[52,53]. Before cell treatments, antibodies were sterilized by filtration using a cellulose acetate membrane filter (0.22 μm, Corning). Antibody concentrations were determined using the Bicinchoninic Acid (BCA) kit and by measuring the absorbance at 280 nm (refs 22,52).

**Surface Plasmon Resonance.** Kinetic interactions between antibodies and Ras proteins were measured at 25 °C using Biacore 2000 surface plasmon resonance (GE Healthcare)[47,52]. RT11 or RT11-i (20 μg ml$^{-1}$) antibodies in Na-acetate (pH 4.5) were immobilized onto the carboxymethylated dextran surface of a CM5 sensor chip at the level of approximately 2,000 response units (RUs). Subsequently, GppNHp- or GDP-loaded GST-fused KRas (WT, G12D, G12V, G13D, and Q61H), GST-fused HRas (WT and G12V), and GST-fused NRas (WT and Q61R) proteins, serially diluted to various concentrations (100–6.25 nM) in running buffer (TBS (50 mM Tris-HCl, pH 7.4, 137 mM NaCl), 5 mM MgCl₂ and 0.01% (v/v) Tween-20), were injected over immobilized RT11 or RT11-i at a flow rate of 30 μl min$^{-1}$ for 3 min with 3 min dissociation per cycle. After each cycle, surfaces were regenerated with buffer (20 mM NaOH, 1 M NaCl, pH 10.0) for 1 min. The binding data were normalized by subtracting the response of a blank cell and then globally fitted using the BIAevaluation software to obtain kinetic interaction parameters.

**ELISA.** Binding specificity of antibodies to antigens was determined by ELISA. The 96-well EIA/RIA plates (Corning) were coated for 1 h at 25 °C with 5 μg ml$^{-1}$ of the indicated antibody and blocked with a blocking solution (TBST (TBS, 0.1% (v/v) Tween-20), 5 mM MgCl₂, 4% (w/v) BSA). After washing with TBST, 100 nM of GST-fused protein (Ras WT or mutants) or His-fused protein (RRas, MRas, or

Rac1), which was loaded with GppNHp or GDP in the blocking solution, was applied to each well for 1 h at 25 °C. After washing with TBST, bound protein was detected by labelling with an HRP-conjugated rabbit anti-GST antibody (Sigma-Aldrich, A7340, dilution 1:4,000) or HRP-conjugated goat anti-His antibody (Sigma-Aldrich, A7058, dilution 1:2,000)) and subsequent incubation with ultra TMB (3,3′,5,5′-tetramethylbenzidine)-ELISA solution (Thermo Scientific, 34028). Absorbance was read at 450 nm on a VersaMax microplate reader (Molecular devices). The results are presented after subtracting the BSA-coated background control value.

**Competition ELISA.** Various concentrations of TMab4 or RT11 (1 μM–5.65 pM) were competed for binding to the respective plate-coated cRaf$_{RBD}$ (20 μg ml$^{-1}$), RalGDS$_{RBD}$ (200 μg ml$^{-1}$), or PI3Kα (p110α/p85α, Signalchem, P27-102H) (10 μg ml$^{-1}$) for 1 h at 25 °C with excess amount of His-KRas$^{G12D}$·GppNHp (1 μM for cRaf$_{RBD}$, 10 μM for RalGDS$_{RBD}$) or biotinylated His-KRas$^{G12D}$·GppNHp (5 μM for PI3Kα). Bound His-KRas$^{G12D}$·GppNHp and biotinylated His-KRas$^{G12D}$·GppNHp were then detected using the HRP-conjugated goat anti-His antibody and HRP-conjugated streptavidin (Thermo Scientific, 21130, dilution 1:10,000), respectively, as described above. The binding data were processed by nonlinear regression analysis using GraphPad PRISM (GraphPad software, Inc.).

**GTPase assay of Ras proteins.** GST-fused KRas (WT and G12D mutant) was loaded with GTP (Sigma-Aldrich, G8877) following the protocol for the preparation of GppNHp-loaded Ras protein. GTPase activity of Ras proteins was measured using an EnzCheck phosphate assay kit (Thermo Scientific, E6646) according to the manufacturer's instructions[27]. Briefly, assay buffer only (intrinsic GTPase activity), indicated antibody (15 μM final concentration) or RasGAP (15 μM final concentration) was added to a reaction mixture (30 μM GTP-loaded KRas protein (WT or G12D), 200 μM 2-amino-6-mercapto-7-methylpurine riboside (MESG), and 50 U ml$^{-1}$ purine nucleoside phosphorylase (PNP) in assay buffer (50 mM Tris-HCl, pH 8.0, 1 mM DTT)) in 96-well half-area microplates (Corning). GTP hydrolysis was initiated by the addition of MgCl₂ (10 mM final concentration). The absorbance at 360 nm was measured at 20 °C for 20 min at every 10 s interval on a MULTISKAN GO plate reader (Thermo Scientific). The phosphate concentration ([Pi]$_t$) at each time point was determined by comparison with a phosphate standard curve and plotted against time. The hydrolysis rate constant ($k$) was determined by fitting the data to a single-phase, exponential non-linear regression curve with the equation [Pi]$_t$ = [Pi]$_0$ + ([Pi]$_{final}$ − [Pi]$_0$) (1 − exp(− $kt$)) in GraphPad Prism software[27].

**Cell lines.** The human cell lines, cervix carcinoma HeLa cells, colorectal carcinoma SW480, LoVo, HT29, and Colo320DM cells, pancreatic carcinoma AsPC-1 and PANC-1 cells, soft tissue sarcoma HT1080 cells, lung carcinoma H1299, acute myeloid leukemia HL60 cells, chronic myeloid leukemia K562 cells, breast carcinoma MCF-7 cells and mouse embryonic fibroblast NIH3T3 cells were purchased from the American Type Culture Collection (ATCC). All cell lines were authenticated by DNA short tandem repeat profiling (ABION CRO, Korea), and used within 20 passages. HeLa, PANC-1, and NIH3T3 cells were maintained in DMEM (HyClone) medium, and SW480, LoVo, HT29, Colo320DM, AsPC-1, HT1080, H1299, HL60, K562 and MCF-7 cells were maintained in RPMI1640 (HyClone) medium. All cells were cultured in growth media that was supplemented with 10% (v/v) heat-inactivated FBS (HyClone), 100 U ml$^{-1}$ penicillin, 100 μg ml$^{-1}$ streptomycin and 0.25 μg ml$^{-1}$ amphotericin B (HyClone). All cell lines were maintained at 37 °C in a humidified 5% CO₂ incubator and routinely screened for *Mycoplasma* contamination (CellSafe).

**Generation of KRas$^{G12V}$-expressing NIH3T3 cells.** The cDNA fragment encoding KRas$^{G12V}$ (residues 1–189) was inserted at the C-terminus of mCherry in pmCherry-C1 plasmid (Clontech)[54] and HA in the pcDNA3.1-HA plasmid[22] to generate the expression plasmid for mCherry-KRas$^{G12V}$ and HA-KRas$^{G12V}$, respectively. NIH3T3 cells were transfected with the expression plasmid using Lipofectamine 2000 (Thermo Scientific, 11668019) and then selected in media containing 0.8 mg ml$^{-1}$ G418 (Sigma-Aldrich, A1720) to isolate the stable cell line expressing mCherry-KRas$^{G12V}$ or HA-KRas$^{G12V}$.

**Cytosolic split GFP complementation assay.** To estimate the cytosolic localization and cytosolic access of antibodies, an enhanced split GFP complementation assay was performed as specified in the figure legend[23,24]. TMab4-GFP11-SBP2, RT11-GFP11-SBP2, TMab4-i-GFP11-SBP2 and RT11-i-GFP11-SBP2 antibodies were prepared by co-expression of the respective GFP11-SBP2-fused HC and its cognate LC in HEK293F cells, as described above.

**Confocal immunofluorescence microscopy.** Cellular internalization and localization of antibodies in cultured cells were detected by confocal microscopy[22,23]. Briefly, cells (5 × 10$^4$) that were grown on 12-mm diameter coverslips in 24-well culture plates were treated with indicated antibodies, as specified in the figure legends. Internalized antibodies were detected with Alexa488-conjugated goat

anti-human IgG antibody (Invitrogen, A11013, dilution 1:500) for 1 h at 25 °C. Ras proteins were detected with rabbit anti-Ras antibody (Abcam, ab108602, dilution 1:100) and subsequently with TRITC-conjugated anti-rabbit antibody (Sigma-Aldrich, T6778, dilution 1:250) for 1 h at 25 °C. The nucleus was stained with Hoechst 33342 in PBS for 5 min at 25 °C. After mounting the coverslips onto glass slides with Perma Fluor aqueous mounting medium (Thermo Scientific, TA-030-FM), center-focused single z-section images were obtained on a Zeiss LSM710 system with ZEN software (Carl Zeiss).

**Western blotting.** Indicated cells ($3 \times 10^5$ cells per well) were seeded in six-well plates, cultured overnight, and then treated with the conditions specified in the figure legends. Western blotting was performed using specific antibodies following the standard procedure[22,53]. For quantification of western blotting data, band intensities were quantified using ImageJ software and normalized to values of the loading control. The phosphorylation levels of proteins were normalized to the total levels of each protein, equivalently loaded on SDS–PAGE gels[55]. Relative band intensity was expressed as a ratio compared to the value of the corresponding control. The original scans of western blots are provided as Supplementary Fig. 12.

**Immunoprecipitation.** To exclude the possibility of IP by antibodies released from endosomes during cell lysis, IP experiments were performed with endosome-depleted cell lysates, which were prepared by removal of early and late endosomes from the cell lysates using density gradient centrifugation[56]. Indicated cells ($1 \times 10^8$ cells per well in 100-mm dish) were treated with PBS buffer or 2 μM of indicated antibodies for 12 h at 37 °C. The cells were washed twice for 30 s at 25 °C with low-pH glycine buffer (200 mM glycine, 150 mM NaCl, pH 2.5) and were then homogenized on ice with homogenization buffer (250 mM sucrose, 3 mM imidazole, pH 7.4, 1 mM EDTA, 10 mM $MgCl_2$ and protease inhibitors (Halt Protease Inhibitor Cocktail, Thermo Scientific, 78440)) and adjusted to 40.6% sucrose concentration by adding 62% sucrose (1:1.2 v/v). Discontinuous sucrose gradients were established in ultracentrifuge tubes with 2 ml homogenates containing 40.6% sucrose, subsequently overlaid with 3 ml of 35% sucrose, 2 ml of 25% sucrose and 5 ml of homogenization buffer. Sucrose gradients were centrifuged in an SW41 swinging-bucket rotor (Beckman) at 100,000g for 3 h at 4 °C. After centrifugation, the endosome-depleted fractions were collected from the bottom of each gradient by puncturing with a needle (26½-gauge). Late endosomes and lysosomes were found on the interface between 25% sucrose and homogenization buffer, whereas early endosomes and carrier vesicles were at the 35–25% interphase. The removal of endosomal fractions from the whole cell lysates was assessed by monitoring Rab5, an early endosome marker[22]. The endosome-depleted cell lysates were mixed and incubated for 30 min on ice with Ras IP buffer (50 mM Tris-HCl, pH 7.4, 150 mM NaCl, 1% NP-40, 10% glycerol, 10 mM $MgCl_2$ and protease inhibitors)[55]. The endosome-depleted cell lysates were then subjected to IP for 2 h at 4 °C with protein A agarose to pull down the antibodies (TMab4, RT11, TMab4-i, and RT11-i), anti-HA antibody (Covance, MMS-101 P), and subsequent incubation with Protein A/G agarose (Santa Cruz, sc-2003) to pull down HA-KRas$^{G12V}$ proteins, or cRaf$_{RBD}$-immobilized agarose beads (Merck Millipore, 14–278) to pull down the GTP-bound active Ras proteins. The complexes are subsequently washed with lysis buffer and equal precipitates were analysed by western blotting with β-actin as a loading control[53,55].

**Cell proliferation assay.** Cells, seeded at a density of $1 \times 10^4$ cells per well in 24-well plates, were cultured for 24 h, and then treated twice, at time 0 and 72 h, with the indicated concentrations of antibodies or the pharmacological inhibitor (sorafenib (Selleckchem, S7397) or LY294002 (Millipore, 440202)) for 6 d, as specified in the figure legend. After the 6 d incubation, cells were collected and viable cells were counted by trypan blue staining. The results are presented as the percentage of viable cells relative to that of the PBS-treated control[53].

**Soft agar colony formation assay.** Cells ($1 \times 10^3$) in 0.5 ml of DMEM or RPMI1640 containing 10% (v/v) FBS and 0.35% (w/v) SeaPlaque agarose were overlaid onto bottom agar consisting of 0.5 ml of DMEM or RPMI1640 containing 10% (v/v) FBS and 0.6% (w/v) agarose in a 12-well culture plate[57]. The cells were incubated at 37 °C and treated with 2 μM of antibodies in growth medium every 72 h for 14–21 d. Following treatment, the cell colonies were stained with BCIP/NBT (5-bromo-4-chloro-3-indolyl phosphate/Nitro blue tetrazolium (Sigma-Aldrich, B5655)), photographed, and analysed with ImageJ software. Colonies with a diameter $> 200$ μm were counted.

**Intracellular KRas$^{G12V}$-cRaf$_{RBD}$ interaction inhibition assay.** The ability of RT11 to interrupt physical interactions between activated KRas and effector proteins in living cells was monitored by analysing the subcellular localization of eGFP-tagged cRaf$_{RBD}$ protein in KRas$^{G12V}$-harbouring SW480 cells[10,58]. To construct a plasmid encoding the eGFP-fused cRaf$_{RBD}$ protein, cRaf$_{RBD}$ (residues 51–220) was inserted at the C-terminus of eGFP in pEGFP-C2 (Clontech). SW480 cells were transfected with the eGFP-cRaf$_{RBD}$-expression plasmid using Lipofectamine 2000 (Invitrogen). After three passages under selection with 3 mg ml$^{-1}$ G418, the cells were treated with antibodies at the indicated doses for

12 h at 37 °C in serum-free media. After two washes with PBS, the cells were fixed with 4% paraformaldehyde in PBS for 10 min at 25 °C. After staining the nucleus with Hoechst 33342 in PBS for 5 min at 25 °C, center-focused single z-section images were obtained by confocal microscopy as described above[23,53].

**Xenograft tumour models.** All animal experiments were approved by the Animal and Ethics Review Committee of Ajou University and performed in accordance with the guidelines established by the Institutional Animal Care and Use Committee[53]. The approval ID for using the animals was No. 2015-0003 by the Animal Core Facility of Ajou University. For xenograft tumour models, 4-week-old, female, BALB/c athymic nude mice (NARA bio, Korea) weighing 17–20 g were inoculated subcutaneously into the right thigh with SW480 ($5 \times 10^6$ cells per mouse), LoVo ($2 \times 10^6$ cells per mouse), HT1080 ($3 \times 10^6$ cells per mouse), or Colo320DM ($5 \times 10^6$ cells per mouse) cells in 150 μl of a 1:1 mixture of PBS/Matrigel (BD Biosciences, 354234). When the mean tumour volume reached approximately 80–100 mm$^3$, mice were randomly assigned to treatment cohorts, and antibodies (TMab4-i and RT11-i) or a PBS vehicle control, as specified in the figure legend, were administered intravenously via the tail vein, in a dose/weight-matched manner. The cetuximab dose for all experiments was 50 mg kg$^{-1}$ intraperitoneally twice weekly. Tumour volumes and body weight were recorded at regular intervals until tumours reached approximately 1,000 mm$^3$, at which time mice were killed. Tumour volume ($V$) was evaluated using digital calipers and estimated by the formula $V = L \times W^2/2$, where $L$ and $W$ are the long and short lengths of the tumour, respectively[53]. Tumour growth inhibition (TGI) by RT11-i compared to that by TMab4-i was determined on the last day of the study according to the formula: TGI (%) $= (100 - (V_f^{RT11-i} - V_i^{RT11-i})/(V_f^{TMab4-i} - V_i^{TMab4-i}) \times 100)$, where $V_i$ is the initial mean tumour volume in the RT11-i or TMab4-i treatment group, and $V_f$ is the final mean tumour volume in the RT11-i or TMab4-i treatment group, as indicated by the superscript notation[44]. Mice were killed with $CO_2$ asphyxiation, and some tumours were collected for histological analysis.

**Immunofluorescence microscopy of tumour tissues.** The extracted tumours from mice were fixed in 4% paraformaldehyde overnight at 4 °C, cryoprotected in 30% sucrose for 24 h, and then frozen in OCT embedding medium (Tissue-Tek). For immunofluorescence staining, 20-μm thick cryosections were prepared and incubated with blocking solution (2% (w/v) BSA in PBS) for 1 h at 25 °C. Tissue sections were incubated with specific primary antibodies overnight at 4 °C, washed three times with PBST (PBS pH 7.4, 0.1% (v/v) Tween-20) for 10 min, and then incubated with appropriate secondary antibodies for 1 h at 25 °C. Apoptotic cells were detected by a standard TUNEL assay using the DeadEnd Fluorometric TUNEL System (Promega). The nuclei were stained with Hoechst 33342 for 5 min at 25 °C. After tissue sections were washed three times with PBST and mounted on slides with Perma Fluor aqueous mounting medium, center-focused single z-section images were obtained on a Zeiss LSM710 system with ZEN software (Carl Zeiss). Using the acquired fluorescence images from each tissue, the fluorescence intensity and number of positively stained cells were quantified using ImageJ software and presented as the relative percent staining compared to that of the PBS-treated control.

**Pharmacokinetic studies in mice.** Female BALB/c nude mice, between 4 and 6 weeks old, received a single injection of 20 mg kg$^{-1}$ of the indicated antibodies in a total volume of 200 μl via the tail vein. In time intervals of 30 min, 1, 4, 8, 12, and 24 h, and 2, 4, 7, 14, 21 and 28 days, blood samples were taken from the tail vein or retro orbital sinus of $CO_2$-anaesthesized mice ($n = 3$ per time point) and incubated for 30 min at 25 °C. Clotted blood was centrifuged at 12,000 r.p.m. for 10 min at 25 °C, and serum samples were stored at –80 °C. Concentrations of human IgG in the serum samples were determined by ELISA. The 96-well EIA/RIA plates (Corning) were coated with 5 μg ml$^{-1}$ of polyclonal goat anti-human IgG (Fab specific) in PBS at 4 °C overnight. The plates were washed with PBST and blocked for 1 h with blocking buffer (PBS, pH 7.4, 1% (w/v) BSA). Samples, as well as purified RT11-i and TMab4-i, used as standard controls, were diluted in blocking buffer and added to the plates with subsequent incubation for 2 h at 25 °C. After washing with PBST, the bound antibody was detected by incubation with HRP-conjugated goat anti-human IgG at a 1:10,000 dilution in blocking buffer for 1 h at 25 °C. After washing with PBST, ultra TMB-ELISA solution was added and absorbance was read at 450 nm on the VersaMax microplate reader (Molecular devices). For comparison, the first value (30 min) was set to 100%. Pharmacokinetic parameters, distribution phase serum half-life ($T_{1/2}\alpha$), and elimination phase serum half-life ($T_{1/2}\beta$) were calculated by two-phasic nonlinear regression analysis using GraphPad Prism[41,53,59]. The serum antibody concentration was also estimated in molar concentration assuming that the total blood volume of nude mice of approximately 20 g (body weight) is $\sim 2$ ml (ref. 60).

**Biodistribution imaging in vivo.** Antibodies were conjugated with DyLight 755 and purified using a DyLight 755 Antibody Labeling Kit (Thermo Scientific, 84538) in accordance with the manufacturer's specifications. BALB/c athymic nude mice were inoculated subcutaneously into the right thigh with SW480 ($5 \times 10^6$ cells per mouse) cells in 150 μl of a 1:1 mixture of PBS/Matrigel. When the mean tumour volume reached $\sim 200$ mm$^3$, DyLight 755-labelled antibodies (20 μg)

were intravenously injected into the mice through the tail vein. Before imaging, mice were anaesthetized with 1.5–2.5% isoflurane (Piramal Critical Care). The whole body distribution profiles of antibodies were quantified by *in vivo* fluorescence using the IVIS Lumina XRMS Series III (Perkin Elmer) at the indicated time intervals. After the final scan, tumour tissues (T) and normal organs (N) were excised and imaged *ex vivo*. To reduce the effects of tissue autofluorescence background, manual spectral unmixing was performed and the fluorescence intensity of an identically sized region of interest was then quantified by radiant efficiency (photons s$^{-1}$ cm$^{-2}$ steradian$^{-1}$ µW$^{-1}$ cm$^{-2}$) using Living Image software (PerkinElmer).

**Statistical analysis.** Data are represented as the mean ± s.d. of triplicate samples from one representative experiment based on at least three independent experiments, unless otherwise specified. Comparisons of data from tests and controls were analysed for statistical significance by a 2-tailed, unpaired Student's *t*-test using MS Excel. A one-way analysis of variance with the Newman-Keuls multiple comparison *post hoc* test was used to determine significance for *in vivo* tumour growth experiments. An animal survival curve and the median survival time were estimated by the Kaplan-Meier method and differences in survival were evaluated by a Gehan-Breslow-Wilcoxon test using GraphPad Prism software. A *P* value of $<0.05$ was considered significant.

**Data reporting.** No statistical method was used to pre-determine sample size, but sample size is similar to sample sizes routinely used in the field. The investigators were not blinded to allocation during experiments. No samples or animals were excluded.

**Data availability.** All data in this study are available within the article and its Supplementary Files or available from the authors on request.

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

## Acknowledgements

This work was supported by grants from the Pioneer Research Center Program (2014M3C1A3051470) and the Global Frontier Project (2013M3A6A4043874) from the National Research Foundation (NRF), funded by the Republic of Korea.

## Author contributions

All authors conceived and designed experiments and analysed data. S.M.S. and D.K.C. generated antibodies and carried out biochemical experiments. D.K.C. and S.W.P. performed antibody affinity maturation and purified and characterized antibodies and recombinant proteins. D.K.C., S.M.S., K.J. and J.S.K. carried out the *in vitro* cellular and mouse xenograft experiments. J.B., K.J. and K.H.S. carried out the colony formation experiments. Y.S.K. supervised the research. S.M.S., D.K.C. and Y.S.K. wrote the manuscript with input from all co-authors.

## Additional information

**Competing interests:** S.M.S., D.K.C. and Y.S.K. are listed as inventors on pending patent applications (PCT/KR2015/007626 and PCT/KR2015/007627) related to technology described in this work. These remaining authors declare no competing financial interests.

