## [Peer Review File · Nature Communications]

Reviewers' comments:

Reviewer #1 (Remarks to the Author):

Kim and coworkers describe the engineering, biochemical, in vitro and animal model analysis of bispecific antibodies that bind selectively to the active form of Ras thus blocking RAS-effector protein interactions. This is a beautiful paper in every respect. The protein engineering is creative (and quite challenging -kudos to the authors for pulling it off) and the in vitro and in vivo data thorough and clean. RT11-i represents a new paradigm in therapeutic antibody discovery and a certainly a great advance in Ras targeting not only for mechanistic purposes but also possibly for therapeutic applications
Publication in NC is highly recommended.

Minor points:

- 1) Why did the authors elect to screen a VH-CH1 library rather than just VH? CH1 is normally intrinsically unfolded prior to associating with CL. Did the authors observe mutations in the CH1 or perhaps used a stabilized mutant CH1 construct?
- 2) Can the authors comment on the stability of RT11-i and of course RT11-imb
- 3) Statistics and repetitions should be provided for the IHC data throughout the MS

Reviewer #2 (Remarks to the Author):

In this manuscript, the Kim lab engineer a cytosol-penetrating antibody that recognizes Ras in its active, GTP-bound, state at its effector-binding domain and prevents Ras association to downstream effectors. The Kim lab has pioneered the design and development of cytosol-penetrating antibodies, such as TMab4, and has technical expertise in antibody engineering with an interest in cytosolic targets. Using yeast surface display and affinity maturation, the authors construct a TMab4 derivative, RT11-i, which retains the light-chain ability to enter the cytosol with new heavy-chain specificity for the Ras effector-binding domain. RT11-i is high affinity (4-17nM) and binds to all isoforms and many oncogenic mutations of Ras to prevent effector binding. The potential to use antibodies which penetrate cells to target difficult intracellular targets is appealing on the one hand, but is plagued by the difficulty of having a 150,000 MW protein penetrate the cell membrane. If this could be done, the potential would be significant. The data however suggest the current form of the antibody is not achieving the stated goals, particularly points 3, 4, 5, 6:

- 1) In Figure 1D, the authors demonstrate by competitive ELISA that RT11 competes with Raf kinase and RalGDS for association to the K-Ras effector binding domain. Is there a reason PI3K was not included? In Figure 3E/F, the authors show dose-dependent p-MEK and p-ERK inhibition with inconsistent p-AKT inhibition. Competitive ELISA on PI3K might explain this phenomenon.
- 2) In Figure 2, the RT11 localizes to the inside of cells using the split GFP reporter, yet, it does not localize to the cell membrane, where Ras is localized. There are several problems with the immunofluorescence. First, the RAS localization is not at the cell membrane, but is generally cytosolic. The typical staining is a fine line at the membrane, since Ras does not cycle to the cytosol. This suggests that both the Ras staining needs optimization, and that the RT11 antibody may not be escaping the endosomes.
- 3) The IPs are problematic because during lysis, the RT11 may be released from endosomes and have access to RAS which then could interact. I'm not confident that this technique is really useful, unless more controls can be included.
- 4) In Figure 3, the dose required to give modest cell killing (10 microM) is quite high, suggesting poor activity, and the need to dose very high in animals. Almost no effect is seen at the 2microM dose. When the signaling is done later in this figure, 5 microM is used, and these effects especially on Akt-P are very modest (and not dose dependent as commented on in the text).
- 5) In Figure 4f, when the RT11-i generation antibody is made, the level of "specific" killing by it,

vs. TMab4-I, the negative control is extremely modest. I even question the difference noted in 4f, for the SW480 cells. I just don't see any really significant difference between the positive and the negative control antibody. This is where the efficacy of the engineered antibody becomes problematic.

6) In Figure 5, the dose of antibody in the blood stream is at the nM level, not a level seen to be active in vitro, so this is a large mismatch between in vitro and in vivo

7) The dosing for efficacy in Figure 6 is done Q2D or Q5D, which is extremely frequent for an antibody therapy, and the doses are extremely high. Yet, the efficacy is very very modest. For such a target, the efficacy of blocking K-Ras mutants should be strong tumor regression.

8) Since the antibody recognizes the GTP-bound state of Ras, the authors should check if the antibody affects GTP hydrolysis to K-RasWT and K-RasG12D in vitro.

9) The authors state that the affinity for H-Ras to the RAF-RBD is 160nM. The authors should include citations on more recent papers on Raf and PI3K binding affinity particularly:

a. Kiel C, Filchtinski D, Spoerner M, Schreiber G, Kalbitzer HR, Herrmann C. Improved Binding of Raf to Ras-GDP Is Correlated with Biological Activity. *The Journal of Biological Chemistry*. 2009;284(46):31893-31902. doi:10.1074/jbc.M109.031153.

b. Hunter, J. C. et al. Biochemical and Structural Analysis of Common Cancer-Associated KRAS Mutations. *Mol. Cancer Res.* 13, 1325–1335 (2015).

10) The authors show that in HeLa cells approximately 1:10 of extracellular concentration of RT11 is retained in the cytosol (Supplementary Table 2). The concentration of Ras in HeLa cells is 0.4-1.6uM (Fujioka, A. et al. *Journal of Biological Chemistry* 281, 8917–8926 (2006)). It is a challenge to inhibit this large quantity of Ras with the current antibody, which has still very poor permeability across the membrane. The authors might want to discuss how the TMabs internalization can be improved through future rounds of antibody engineering.

Overall, the data do not support the authors' strong conclusions about the ability to target K-Ras intracellularly.

Reviewer #3 (Remarks to the Author):

The manuscript from Shin and colleagues describes the generation of an antibody (RT11) targeting RAS proteins with a quite beautiful specificity for RAS mutant tumours. The authors characterize the effect of such antibody in in vivo and in vitro assays, showing decreased tumour proliferation of cells in both experimental settings. They also showed how this antibody impairs downstream signalling of the two major RAS effector pathways: ERK and PI3K.

Although the results shown are potentially interesting for the field, there are a few issues that are not clearly addressed in the manuscript at the moment and would be important to address:

- The authors claim through the text that their antibody is specific for KRas. However in figure 1c they show it also binds to HRas and NRas and they showed in figure 3a and supplementary Fig. 4a that RT11 inhibits proliferation of NRas mutated cell lines. So it is not specific for KRas. This should be removed from the text.

- Since the antibody was generated against Ras-GTP and it can recognize HRas, NRas and KRas, authors should check how specific the antibody is and if it still can recognize and bind other Ras-GTP superfamily members such as MRas, RRas, or even Rac.

- In figure 2b the authors showed co-localization of RT11 with mutant KRas at the inner plasma membrane. Although localization of their antibody seems different from their control, images are very small and it is very difficult to determine proper localization of antibodies. Staining looks all over the cytoplasm and not inner plasma membrane as authors claim. Images with a higher magnification are required and/or images with enlarged areas where results can be easily observed. Also arrows indicating areas with the results they want to show will help readers to

quickly spot differences.

- Soft agar colony formation assays with other Ras mutations should be shown to determine specificity of oncogenic KRas recognition.
- In figure 3A authors show decreased proliferation of cells treated with their RT11 antibody. In order to evaluate the proliferation effect of the antibody with other conventional therapies it would be nice to compare results with Akt and/or ERK inhibitors.
- In figure 3e and f the effect of the antibody on Ras downstream pathway inhibition seems to have a strong variation. Cells have been treated with different growth factors and one wonders if the differences are because of the use of FBS or EGF. On NIH3T3-HAKRasG12V cells decrease in p-ERK and p-Akt seems minimal. Downstream targets of pAkt specially need to be explored in order to determine downregulation of the pathway.
- On figure 4d images are very low magnification to properly see inner membrane localization of the antibodies. RT-11-i and TMab4-i seems to be localized in the same places. Localization of the antibodies on Ras WT cells nice would be good.
- On figure 4f the difference on proliferation between RT-11-i and TMab4-i is very small, especially if we compared the difference in proliferation of both antibodies before targeting them with the integrin sequence. Authors should show if proliferation decrease is increased with higher antibody concentrations as done in figure 3a.
- On figure 5b it would be good to see how the RT11 antibody localises compared to the RT11-i.
- In their xenografts experiments it would be good to see effect of the TR11-i in for example NRas mutant tumours and compare it with KRas.
- In figure 6c authors claim only RT11-i co-localized with KRas at the inner membrane. This is not possible to see on the provided images. As before, higher magnification images should be shown.
- In discussion authors claim that integrin-bound RT antibody "... did not compromise the inherent biochemical and biological properties of RT11". I don't agree with this since localization and proliferation seems to be different. Also the integrin signal itself has effect on phosphorylation of downstream targets and also on some xenografts. This should be discussed.

Minor points:

- On page 18, on the first paragraph: "Importantly, RT11-i markedly inhibited the tumor growth of the three tumor types carrying oncogenic KRas..." Authors present only two tumours carrying KRas mutations. This should be corrected.

Responses to Reviewer comments for MS# NCOMMS-16-20913A

We greatly appreciate the valuable comments from the reviewers, which led to changes that substantially improved the manuscript. In response to comments from the three reviewers, we provide additional experimental data to clarify some results with appropriate explanations. We also added/edited some sentences to address their concerns in the revised manuscript. All of these changes are highlighted in *red color* in the revised manuscript and below in the responses to the reviewer comments.

Responses to specific comments from reviewer #1

General comments: Kim and coworkers describe the engineering, biochemical, in vitro and animal model analysis of bispecific antibodies that bind selectively to the active form of Ras thus blocking Ras-effector protein interactions. This is a beautiful paper in every respect. The protein engineering is creative (and quite challenging - kudos to the authors for pulling it off) and the in vitro and in vivo data thorough and clean. RT11-i represents a new paradigm in therapeutic antibody discovery and a certainly a great advance in Ras targeting not only for mechanistic purposes but also possibly for therapeutic applications. Publication in NC is highly recommended.

Comment #1-1) Why did the authors elect to screen a VH-CH1 library rather than just VH? CH1 is normally intrinsically unfolded prior to associating with CL. Did the authors observe mutations in the CH1 or perhaps used a stabilized mutant CH1 construct ?

Response: Thank you for your comment. As we wrote in the original manuscript, we screened a human VH-CH1 (HC) library, rather than just a VH library, to isolate active Ras specific HC binder in the Fab format after mating with a fixed LC (VL-CL) of TMab4. The reason is that, compared with scFv (VH-linker-VL), Fab is more reliably converted into an IgG format without the loss of antigen-binding properties.

For VH-CH1 library construction, we did not employ a stabilized mutant CH1, but rather used wild-type human IgG1 CH1. As you can see in the Supplementary

Fig. 1a of the original manuscript, the VH-CH1 library was well-expressed on the yeast cell surface, indicating that the VH-CH1 library is properly folded as it was displayed on the eukaryotic yeast cell surface; this was most likely due to the eukaryotic quality control system, consistent with a previous report [Stavenhagen JB et al. (2007) *Cancer Res.* 67(18):8882-90]. Sequencing of numerous HC clones before and after screening did not reveal any mutation in the CH1 region.

Comment #1-2) Can the authors comment on the stability of RT11-i and of course RT11 iMab.

Response: In response to the reviewer’s comment, we evaluated the thermal stability of RT11 and RT11-i by incubating them at 50 °C for various time periods (6, 12, 24, and 48 h) and then measuring the selective antigen binding activity with ELISA and colloidal stability with size exclusion chromatography. Even after 48 h of incubation at 50 °C, as shown below, both RT11-i and RT11 retained their selective binding to the GppNHp-bound active KRas^{G12D} form over the GDP-bound inactive form without formation of soluble oligomers, compared with those incubated at 4 °C, indicating that RT11-i possesses comparable thermal stability to that of RT11.

This new data was added in Supplementary Fig. 8 and described in the Result of the revised manuscript, as described below.

[Supplementary Fig. 8] and its legend (p. 11) in the revised manuscript,

"Supplementary Figure 8. RT11-i possesses comparable thermal stability to that of

RT11. (a,b) Thermal stability of RT11 and RT11-i was assessed by ELISA for selective antigen binding **(a)** and size exclusion chromatography **(b)** after incubating the antibodies (2 mg/ml) at 50 °C for the indicated periods. In **(a)**, the binding activity (%) was relative to the initial binding of the antibody stored at 4 °C. Error bars represent the mean \pm s.d. ($n = 3$). In **(b)**, the size exclusion chromatogram of antibodies (20 μ l of 1 mg/ml) was monitored at 280 nm. Antibody stored at 4 °C was included as a control. Two independent analyses were performed with the same results. The arrows indicate the elution positions of molecular weight standards."

In the **[Results]** (p.11) of the revised manuscript

"Even after 48 h of incubation at 50 °C, both RT11 and RT11-i maintained selective binding to active KRas^{G12D} form without the formation of soluble oligomers (Supplementary Fig. 8), indicating that RT11-i possesses comparable thermal stability to that of RT11. Thus, the fusion of the RGD10 peptide to the N-terminus of the LC of RT11 did not compromise the biochemical and biophysical properties of RT11."

Comment #1-3) Statistics and repetitions should be provided for the IHC data throughout the MS.

Response: In response to the reviewer's comment, we stated the number of replicates for the IHC experiments using tumor samples in the revised Figure legend (Fig. 6b-c, Fig. 7d, and Supplementary Fig. 11a-b). We performed statistical analysis using a one-way ANOVA followed by the Newman-Keuls post-test and described it in each Figure legend and in the Methods.

In the legend of **[Fig. 6b,c]** of the revised manuscript,

"**(b,c)** Immunohistochemical images showing the levels of p-ERK1/2 and p-Akt **(b)** or cellular penetration and co-localization of antibodies with the active Ras form **(c)** in SW480 tumor tissues excised from mice following treatment described in **(a)**. **Images are representative of three independent experiments.** Nuclei were counterstained with Hoechst33342 (blue). Scale bar, 100 μ m in **(b)** or 10 μ m in **(c)**. In **(b)**, the right panel shows

the percent relative fluorescence intensity compared to that of vehicle-treated control. Error bars, \pm s.d. of five random fields for each tumor (two tumors per group). In (c), the areas in the white boxes are shown at a higher magnification for better visualization. The arrows indicate the co-localization of RT11-i with activated Ras. In (a,b), statistical analysis was performed using a one-way ANOVA followed by the Newman-Keuls post-test. $*P < 0.05$, $**P < 0.01$, $***P < 0.001$ vs. TMab4-i; n.s., not significant."

In the legend of [Fig. 7d] of the revised manuscript,

"Images are representative of at least two independent experiments. The right panel shows the percent relative fluorescence intensity compared to that of the vehicle-treated control group. Error bars, \pm s.d. of five random fields for each tumor (two tumors per group). Statistical analysis was performed using a one-way ANOVA followed by the Newman-Keuls post-test. $*P < 0.05$, $**P < 0.01$, $***P < 0.001$; n.s., not significant."

In the legend of [Supplementary Fig. 11] of the revised manuscript,

"Error bars represent the mean \pm s.d. of five random fields for each immunofluorescence sample ($n =$ two tumors per group). In (a, b, d and e), nuclei were counterstained with Hoechst 33342 (blue). In (b,d and e), statistical analyses were performed using a one-way ANOVA followed by the Newman-Keuls post-test. $*P < 0.05$, $**P < 0.01$, $***P < 0.001$ vs. TMab4-i; n.s., not significant. Images are representative of at least two independent experiments."

Responses to specific comments from reviewer #2

General comments: In this manuscript, the Kim lab engineer a cytosol-penetrating antibody that recognizes Ras in its active, GTP-bound, state at its effector-binding domain and prevents Ras association to downstream effectors. The Kim lab has pioneered the design and development of cytosol-penetrating antibodies, such as TMab4, and has technical expertise in antibody engineering with an interest in cytosolic targets. Using yeast surface display and affinity maturation, the authors construct a TMab4 derivative, RT11-i, which retains the light-chain ability to enter the cytosol with new heavy-chain specificity for the Ras effector-binding domain. RT11-i is high affinity (4-17nM) and binds to all isoforms and many oncogenic mutations of Ras to prevent effector binding. The potential to use antibodies which penetrate cells to target difficult intracellular targets is appealing on the one hand, but is plagued by the difficulty of having a 150,000 MW protein penetrate the cell membrane. If this could be done, the potential would be significant. The data however suggest the current form of the antibody is not achieving the stated goals, particularly points 3, 4, 5, 6:

Comment #2-1) In Figure 1D, the authors demonstrate by competitive ELISA that RT11 competes with Raf kinase and RaIGDS for association to the K-Ras effector binding domain. Is there a reason PI3K was not included? In Figure 3E/F, the authors show dose-dependent p-MEK and p-ERK inhibition with inconsistent p-AKT inhibition. Competitive ELISA on PI3K might explain this phenomenon.

Response: Thank you for your comment. Even though we tried to express and purify the Ras-binding domain (RBD) of PI3K (PI3K_{RBD}) in *E. coli*, the yield and purity were too low to perform competitive ELISA. To address the reviewer's comment, we purchased recombinant PI3K α (p110 α /p85 α) protein (SignalChem) for the competition ELISA. As shown below, RT11 efficiently competed with PI3K for binding to KRas^{G12D}-GppNHp, which is expected based on the results that RT11 binds to the protein-protein interaction (PPI) interface between active Ras and effector proteins, including PI3K (Fig. 1d,e and Fig. 3). Thus RT11 can inhibit the downstream

signaling mediated by active Ras-GTP-effector PPIs, such as Raf-MEK1/2-ERK1/2 and PI3K-Akt.

Regarding the reviewer's concern of inconsistent p-Akt inhibition in Fig. 3e,f, quantification of band intensities of the Western blotting data revealed that RT11 dose-dependently attenuated the activation of MEK1/2, ERK1/2, and Akt. The quantification of the original results was shown in the response to comment #2-4. To address the reviewer's concern, we performed the experiments shown in Fig. 3e,f with higher RT11 concentrations of 2 and 10 μ M instead of 1 and 5 μ M (used in the original manuscript). The results (shown in the response to comment #2-4) clearly demonstrate the dose-dependent inhibitory effect of RT11 on the downstream signaling of Ras-effector PPIs in HA-KRas^{G12V}-transformed NIH3T3 cells and SW480 cells. Overall, the above results confirm our original claim that competitive blocking of active Ras-effector PPIs by RT11 results in the attenuation of downstream signaling.

The data for the competition ELISA with PI3K was added to Fig. 1d with an appropriate description in the revised manuscript, as shown below.

[Fig. 1d] and its legend in the revised manuscript

"(d) Competition ELISA showing that RT11, but not TMab4, blocks interactions between KRas^{G12D}·GppNHp and effector proteins, such as cRaf_{RBD}, RalGDS_{RBD}, and PI3K α (p110 α /p85 α)."

In the **[Results]** (p.6) of the revised manuscript,

"RT11 efficiently competed with the RBDs of cRaf (cRaf_{RBD}) and RalGDS (RalGDS_{RBD}), as well as PI3K α (p110 α /p85 α) protein for binding to KRas^{G12D}·GppNHp (Fig. 1d), suggesting that RT11 binds to the PPI interfaces between the active Ras and the effector proteins."

In **[Supplementary Methods]** (pp. 29-30) of the revised manuscript,

"Competition ELISA with cRaf_{RBD}, RalGDS_{RBD}, and PI3K α

For competition ELISA, various concentrations of TMab4 or RT11 (1 μ M~5.65 pM) were used for binding to plate-coated cRaf_{RBD} (20 μ g/ml), RalGDS_{RBD} (200 μ g/ml), or PI3K α (p110 α /p85 α) (10 μ g/ml) for 1 h at 25 °C with excess His-KRas^{G12D}·GppNHp (1 μ M for cRaf_{RBD}, 10 μ M for RalGDS_{RBD}) or biotinylated Avi-His-KRas^{G12D}·GppNHp (5 μ M for PI3K α). Bound His-KRas^{G12D}·GppNHp and biotinylated Avi-His-KRas^{G12D}·GppNHp were then detected using an HRP-conjugated goat anti-His antibody (Sigma-Aldrich) and HRP-conjugated streptavidin (Thermo Scientific), respectively, as described earlier. The binding data were processed by nonlinear regression analysis using GraphPad PRISM (GraphPad software, Inc.)."

Comment #2-2) In Figure 2, the RT11 localizes to the inside of cells using the split GFP reporter, yet, it does not localize to the cell membrane, where Ras is localized. There are several problems with the immunofluorescence. First, the RAS localization is not at the cell membrane, but is generally cytosolic. The typical staining is a fine line at the membrane, since Ras does not cycle to the cytosol. This suggests that both the Ras staining needs optimization, and that the RT11 antibody may not be escaping the endosomes.

Response: In Figure 2a, we performed the split-GFP complementation cellular assay in wild-type Ras-expressing HeLa cells. Since Ras^{WT}-harboring HeLa cells have predominantly in the GDP-bound inactive form rather than the GTP-bound active form in the steady states [Hayes TK and Der CJ (2013) Cancer Discov

3(1):24-26; Hirasawa K et al. (2002) Cancer Research 62:1696-1701], RT11-GFP11-SBP2 did not localize to the inner plasma membrane, but localized in the cytosol to assemble with cytosolically-expressed SA-GFP1-10, resulting in a dominant complemented GFP signal in the cytosolic region. This type of RT11 localization was observed in Ras^{WT}-harboring HT-29 cells, shown in the revised Fig. 2b (shown below).

In response to the reviewer comment, we indicated that HeLa cells express Ras^{WT} proteins in the revised manuscript,

In the **[Results]** (p. 7) of the revised manuscript,

"RT11-GFP11-SBP2 incubated with Ras^{WT}-harboring HeLa-SA-GFP1-10 cells exhibited complemented GFP fluorescence in the cytosol, similar to that observed with TMab4-GFP11-SBP2²³, demonstrating that RT11 retains the cytosol-penetrating ability of TMab4 VL."

The reviewer also commented on the Ras staining/localization in mCherry-KRas^{G12V}-transformed NIH3T3 and KRas^{G12V}-harboring SW480 cells, shown in Fig. 2b. In NIH3T3-mCherry-KRas^{G12V} cells, ectopic overexpression of mCherry-fused KRas^{G12V} appeared to result in cytosolic localization of KRas^{G12V} in addition to plasma membrane, consistent with a previous report [Lee HW et al. (2012) Nature Communications 4:1505]. In SW480 cells, we stained for Ras using an anti-Ras antibody (Abcam, ab108602). Since the antibody labels all isoforms of Ras regardless of GTP- or GDP-bound status, wild-type HRas and NRas in the cytosolic region and/or Golgi apparatus can yield cytosolic Ras staining [Ehrhardt A et al. (2004) Mol. Cell. Biol. 24(14):6311–6323] in addition to KRas^{G12V} staining at the plasma membrane.

To clarify the co-localization of RT11 with activated Ras, we repeated the experiments shown in Fig. 2b and produced high-quality images with enlarged areas to clearly show the co-localization of RT11 with activated KRas around the plasma membrane. We further characterized RT11 localization in Ras^{WT}-harboring HT-29 cells as a control. As shown below, RT11 dominantly localized to the cytosol rather than the plasma membrane. These results validated the binding specificity of RT11 in

the intracellular space and confirmed our claim in the original manuscript.

[Fig. 2b] and its legend of the revised manuscript:

"(b) Cellular internalization and co-localization of RT11 (green), but not TMab4 (green), with the inner plasma membrane-anchored active Ras (red) in mCherry-KRas^{G12V}-transformed NIH3T3 and KRas^{G12V}-harboring SW480 cells, analyzed by confocal microscopy. The Ras^{WT}-harboring HT29 cells were also analyzed as a control. The areas in the white boxes are shown at a higher magnification for better visualization. The arrow indicates the co-localization of RT11 with activated Ras. Scale bar, 5 μm."

In the [Results] (pp. 7-8) of the revised manuscript,

"Furthermore, when incubated with mCherry-KRas^{G12V}-transformed mouse fibroblast NIH3T3 and KRas^{G12V}-harboring human colorectal SW480 cells under normal cell culture conditions, internalized RT11, but not TMab4, co-localized with KRas^{G12V} at the inner plasma membrane (Fig. 2b), where active KRas^{G12V}·GTP is anchored^{1,2}. However, in Ras^{WT}-

harboring human colorectal HT29 cells, RT11 predominantly localized throughout the cytosol without enrichment at the plasma membrane (Fig. 2b), similar to TMab4. This could be ascribed to the predominant presence of Ras^{WT} in the inactive form in resting cells^{34,35}."

Comment #2-3) The IPs are problematic because during lysis, the RT11 may be released from endosomes and have access to RAS which then could interact. I'm not confident that this technique is really useful, unless more controls can be included.

Response: To exclude the possibility of IP by antibody released from endosomes during cell lysis, we performed all IP experiments with endosome-depleted cell lysates, which were prepared by eliminating early and late endosomes from whole cell lysates using density gradient centrifugation [de Araújo ME et al, Cold Spring Harb Protoc. 2015; 11:1013-1016, de Araújo ME et al, Methods Mol Biol. 2008;424:317-31]. We verified the elimination of the endosomal fraction from the cellular lysates by probing for Rab5, an early endosome marker. We showed that, in RT11-treated KRas^{G12V}-transformed NIH3T3 cells and KRas^{G12V}-harboring SW480 cells (Fig. 2c), pull-down of RT11 from the endosome-depleted cellular fractions resulted in co-precipitation of KRas^{G12V}, whereas TMab4 failed to. Likewise, we performed IPs with the endosome-depleted cell lysates shown in Fig. 3c,d and Fig. 4e. All of the results demonstrated the physical interaction of RT11/RT11-i with active Ras mutants after cellular internalization and subsequent cytosolic localization.

We replaced the original data from Fig. 2c, Fig. 3c,d and Fig. 4e with the newly performed IP data using endosome-depleted cell lysates of antibody-treated cells in the revised manuscript.

[Fig. 2c] and its legend in the revised manuscript,

"(c) Immunoprecipitation (IP) of KRas mutant with RT11, but not TMab4, from **endosome-depleted cell lysates** of HA-KRas^{G12V}-transformed NIH3T3 and SW480 cells, treated with 2 μ M of antibody for 12 h before analysis. **The endosome-depleted cell lysates were assessed by the absence of Rab5, an early endosome marker²⁴.**"

[Fig. 3c,d] and its legend in the revised manuscript,

"(c,d) IP of endogenous Raf proteins (bRaf and cRaf) with HA-tagged KRas^{G12V} from the **endosome-depleted** cell lysates of HA-KRas^{G12V}-transformed NIH3T3 cells (c) and IP of endogenous KRas^{G12V} with cRaf_{RBD} from the **endosome-depleted cell lysates** of SW480 cells (d)."

[Fig. 4e] and its legend in the revised manuscript,

"(e) IP of endogenous KRas^{G12V} with RT11 or RT11-i, but not by TMab4 and TMab4-i, from endosome-depleted cell lysates of SW480 cells."

In the [Results] (p. 8) of the revised manuscript,

"Immunoprecipitation (IP) using endosome-depleted cell lysates of RT11-treated KRas^{G12V}-transformed NIH3T3 cells and KRas^{G12V}-harboring SW480 cells, using RT11 itself, revealed the physical interaction of RT11 with cytosolic KRas mutants after internalization (Fig. 2c)."

In "Immunoprecipitation (IP)" section of [Supplementary methods] (pp. 32-33) of the revised manuscript,

"To exclude the possibility of IP by antibodies released from endosomes during cell lysis, IP experiments were performed with endosome-depleted cell lysates, which were prepared by removal of early and late endosomes from the cell lysates using density gradient centrifugation, as described previously^{80,81}. Briefly, indicated cells (1×10^8 cells/well in a 100-mm dish) were treated with PBS buffer or 2 μ M of indicated antibodies for 12 h at 37 °C. The cells were washed twice for 30 s at 25 °C with low-pH glycine buffer (200 mM glycine, 150 mM NaCl, pH 2.5) and were then homogenized on ice with homogenization buffer (250 mM sucrose, 3 mM imidazole, pH 7.4, 1 mM EDTA, 10mM MgCl₂ and protease inhibitors) (Halt Protease Inhibitor Cocktail, Thermo Scientific, 78440)) and adjusted to 40.6% sucrose

concentration by adding 62% sucrose (1:1.2 v/v). Discontinuous sucrose gradients were established in ultracentrifuge tubes with 2 ml homogenates containing 40.6% sucrose, subsequently overlaid with 3 ml of 35% sucrose, 2 ml of 25% sucrose and 5 ml of homogenization buffer. Sucrose gradients were centrifuged in an SW41 swinging-bucket rotor (Beckman) at $100,000 \times g$ for 3 h at 4 °C. After centrifugation, the endosome-depleted fractions were collected from the bottom of each gradient by puncturing with a needle (26½-gauge). Late endosomes and lysosomes were found on the interface between 25% sucrose and homogenization buffer, whereas early endosomes and carrier vesicles were at the 35%/25% interphase. The removal of endosomal fractions from the whole cell lysates was assessed by monitoring Rab5, an early endosome marker⁶⁵. The endosome-depleted cell lysates were mixed and incubated for 30 min on ice with Ras IP buffer (50 mM Tris-HCl, pH 7.4, 150 mM NaCl, 1% NP-40, 10% glycerol, 10 mM MgCl₂, and protease inhibitors). The endosome-depleted cell lysates were then subjected to IP for 2 h at 4 °C with protein A agarose to pull down the antibodies (TMab4, RT11, TMab4-i, and RT11-i), anti-HA antibody (Covance, MMS-101P), and subsequent incubation with Protein A/G agarose (Santa Cruz, sc-2003) to pull down HA-KRas^{G12V} proteins, or cRaf_{RBD}-immobilized agarose beads (Merck Millipore, 14-278) to pull down the GTP-bound active Ras proteins. The complexes were subsequently washed with lysis buffer and equal precipitates were analyzed by western blotting with β-actin as a loading control, as described previously⁷⁷."

Comment #2-4) In Figure 3, the dose required to give modest cell killing (10 microM) is quite high, suggesting poor activity, and the need to dose very high in animals. Almost no effect is seen at the 2 microM dose. When the signaling is done later in this figure, 5 microM is used, and these effects especially on Akt-P are very modest (and not dose dependent as commented on in the text).

Response: We understand the reviewer's concerns. Although RT11 did not show very potent activity, it exhibited statistically significant and dose-dependent anti-proliferative activity for oncogenic Ras mutant-harboring tumor cells showing ~14–35% growth inhibition at 2 μM and ~46–53% growth inhibition at 10 μM compared to the control TMab4 (as shown with *P* value in Fig. 3a in the original manuscript). However,

in soft agar colony formation, treatment with RT11 at 2 μ M resulted in ~44–64% suppression in the colony formation of oncogenic Ras mutant tumors compared to treatment with TMab4 (Supplementary Fig. 5c); this represents a more dramatic effect when compared to the results of plate-based proliferation assays. This observation is in line with previous reports demonstrating that oncogenic Ras mutant cancers are more dependent on Ras-driven signaling during anchorage-independent culture conditions (3D soft agar colony formation assay) than during monolayer culture conditions (2D plate proliferation assay) [Fujita-Sato S et al. (2015) *Cancer Res.* 75(14): 2851–2862; Shi XH et al. (2009) *Cancer Gene Therapy* 16:227-236; Fujita M et al. (1999) *Melanoma Res.* 9(3):279-291].

To clearly show the anti-tumor effect of RT11, we further assessed the anti-proliferative activity of RT11 under anchorage-independent growth conditions using soft agar colony formation assays with diverse oncogenic Ras mutant tumor cells, such as LoVo (KRas^{G13D}), HT1080 (NRas^{Q61K}), and H1299 (NRas^{Q61K}) by treatment with RT11 (2 μ M) or TMab4 (2 μ M) every 72 h for 2–3 weeks. As shown below, RT11 resulted in ~44–64% inhibition in colony formation with oncogenic Ras mutant tumor cells, but did not exhibit a significant effect on Ras^{WT} cells, when compared to TMab4 treatment. These results clearly demonstrate that RT11 inhibits the growth of tumor cells harboring oncogenic Ras mutants, but not those with Ras^{WT}.

The above results were added in [Supplementary Fig. 5c] and its legend of the revised manuscript,

"(c) Inhibition of soft agar colony formation of tumor cells by RT11. Anchorage-independent

cell growth was examined by soft agar colony formation assays using the oncogenic KRas mutant, **NRas mutant**, and Ras^{WT} cells, treated with RT11 (2 μM) and TMab4 (2 μM) every 72 h for 2–3 weeks. Following treatment, the number of colonies (**diameter > 200 μm**) was counted after BCIP/NBT staining, as shown in the pictures of the representative soft agar plates (right)."

In the **[Results]** (p. 9) of the revised manuscript,

" Oncogenic Ras mutations are known to drive anchorage-independent tumor growth, another important hallmark of cellular transformation^{21,36}. Based on soft agar colony formation assays, **RT11 treatment resulted in ~44–64% inhibition of anchorage-independent proliferation for oncogenic KRas mutant cells (SW480, LoVo, and PANC-1) and NRas mutant cells (HT1080 and H1229), but not in Ras^{WT} Colo320DM and K562 cells, compared to that after TMab4 treatment (Supplementary Fig. 5c)**. This result demonstrates that RT11 suppresses the tumorigenic activity of **oncogenic Ras mutants**. **Of note, the effect on tumor growth after blocking oncogenic Ras with RT11 was more evident in anchorage-independent growth conditions (soft agar) than that in monolayer culture conditions (Supplementary Fig. 5a,c), which is line with previous observations with siRNA-mediated knockdown of oncogenic Ras^{37,38}.**"

Regarding to the reviewer's concern about inconsistent p-Akt inhibition in Fig. 3e,f, quantification of band intensities of the Western blotting data revealed that RT11 attenuated the activation of MEK1/2, ERK1/2, and Akt in a dose-dependent manner. The quantification data of the original results is shown below.

Quantification of Fig. 3e,f Western blotting data in the original manuscript.

However, to address the reviewer's concern, we repeated the experiments shown Fig. 3e,f with higher RT11 concentrations (2 and 10 μM) than those (1 and 5 μM) used in the original manuscript to clearly show the dose-dependent inhibition of RT11 on downstream Ras signaling. As shown below, RT11 exhibited dose-dependent inhibition of serum- and epidermal growth factor (EGF)-stimulated phosphorylation of MEK1/2, ERK1/2, and Akt in HA-KRas^{G12V}-transformed NIH3T3 cells (Fig. 3e) and SW480 cells (Fig. 3f), respectively. Furthermore, in response to reviewer comment #3-6, we also monitored the activation of p70S6K, a downstream kinase of the PI3K-Akt-mTOR pathway [Chung J et al. (1994) Nature 370:71-75]. As shown below, RT11 also exhibited dose-dependent inhibition of p70S6K in both cell lines (Fig. 3e,f). These results confirmed that RT11 attenuated downstream signaling mediated by Ras-GTP-effector PPIs, such as Raf-MEK1/2-ERK1/2 and PI3K-Akt-mTOR pathways in a dose-dependent manner.

We replaced the original figure with the new data in [Fig. 3e,f] in the revised manuscript.

" (e,f) Inhibitory effect of RT11 on the downstream signaling of KRas-effector PPIs in HA-KRas^{G12V}-transformed NIH3T3 cells (e) and SW480 cells (f), analyzed by western blotting. The cells were serum-starved for 6 h before treatment with antibody, Raf kinase inhibitor sorafenib, or PI3K-Akt inhibitor LY294002 for 6 h in serum-free growth medium. Cells were washed and then stimulated with 10% FBS (e) and EGF (50 ng/ml in serum free-media) (f) for 10 min before cell lysis. The number below the panel indicates relative value of band intensity of phosphorylated proteins compared to that in the PBS-treated control after normalization to the band intensity of respective total protein for each sample. * $P < 0.05$, ** $P < 0.01$, *** $P < 0.001$ vs. PBS-treated control cells. In (b-f), images are representative of at least two independent experiments."

In the [Results] (p. 10) of the revised manuscript,

"We next investigated the effect of RT11 on downstream signaling mediated by Ras·GTP-effector PPIs such as the Raf-MEK1/2-ERK1/2 and PI3K-Akt-mTOR pathways^{1,2}. RT11 exhibited dose-dependent inhibition of downstream signaling mediated by PPIs of Ras-Raf (MEK1/2 and ERK1/2) and Ras-PI3K (Akt and p70S6K) in serum-stimulated HA-KRas^{G12V}-transformed NIH3T3 cells (Fig. 3e) and epidermal growth factor (EGF)-stimulated SW480 cells (Fig. 3f). The Raf inhibitor sorafenib and PI3K-Akt inhibitor LY294002 attenuated only targeted signaling in SW480 cells (Fig. 3f)."

In **[Supplementary Methods]** (pp. 31-32) of the revised manuscript,

"Western blotting

Equal amounts of lysates were analyzed by western blotting with β -actin serving as a loading control. Proteins were visualized using a PowerOpti-ECL western blotting detection reagent (Animal Genetics) and an ImageQuant LAS 4000 mini (GE Healthcare). For quantification of western blotting data, band intensities were quantified using ImageJ software and normalized to values of the loading control⁷⁹. The phosphorylation levels of proteins were normalized to the total levels of each protein, equivalently loaded on SDS-PAGE gels. Relative band intensity was expressed as a ratio compared to the value of the corresponding control."

Comment #2-5) In Figure 4f, when the RT11-i generation antibody is made, the level of "specific" killing by it, vs. TMab4-i, the negative control is extremely modest. I even question the difference noted in 4f, for the SW480 cells. I just don't see any really significant difference between the positive and the negative control antibody. This is where the efficacy of the engineered antibody becomes problematic.

Response: As the reviewer commented, in Fig. 4f, anti-proliferative activity of RT11-i was not very dramatic, when compared to that of TMab4-i. This is mainly due to the anti-proliferative activity of the control TMab4-i. TMab4-i inhibits cell growth by an RGD10-mediated integrin-blocking effect on integrin-mediated cell adhesion, particularly in cells overexpressing integrin $\alpha v \beta 5$, such as SW480 cells (Fig. 4b), under monolayer culture conditions [Desgrosellier JS et al. (2010) *Nature reviews Cancer* 10:9-22; Goodman SL and Picard M (2012) *Trends in Pharmacological Sciences* 33(7):405-412; Howe A et al. (1999) *Current Opinion in Cell Biology*, 10:220-231].

To address the reviewer's concern, the anti-tumor activity of RT11-i was further assessed and compared to that of TMab4-i using soft-agar colony forming assays. For this, we expected more obvious oncogenic Ras-blocking effect, while minimizing the RGD10-mediated integrin-blocking effect, when compared to

monolayer culture conditions [Fujita-Sato S et al. (2015) *Cancer Res.* 75(14): 2851–2862; Shi XH et al. (2009) *Cancer Gene Therapy* 16:227-236; Fujita M et al. (1999) *Melanoma Res.* 9(3):279-291]. We tested diverse oncogenic Ras mutant cells, such as SW480 (KRas^{G12V}), LoVo (KRas^{G13D}), AsPC-1(KRas^{G12D}), PANC-1 (KRas^{G12D}), HT1080 (NRas^{Q61K}), and H1299 (NRas^{Q61K}) cells, using Colo320DM (Ras^{WT}) cells as a control. As shown below, RT11-i exerted an approximate 40–67% suppression of colony formation with oncogenic Ras tumor cells, but not with Ras^{WT} Colo320DM cells, when compared to that using T Mab4-i. These data clearly demonstrate that RT11-i suppresses proliferation of cancer cells by blocking the oncogenic Ras signaling.

The data obtained by performing soft-agar colony forming assays were added to Fig. 4f and the original data shown in Fig. 4f were replaced in Supplementary Fig. 7e, which was described appropriately in the revised manuscript.

[Fig. 4f] and its legend in the revised manuscript,

"(f) Inhibition of tumor cell soft agar colony formation by RT11-i compared to that with T Mab4-i. Following treatment of cells with PBS, T Mab4-i (2 μM), or RT11-i (2 μM) every 72 h for 2–3 weeks, the number of colonies (diameter >200 μm) was counted by BCIP/NBT staining, as shown in the pictures of representative soft agar plates (Supplementary Fig. 7f). The results are presented as percentages compared to the PBS-treated control. Error bars represent the mean ± s.d. (n = 3). **P < 0.01, ***P < 0.001; n.s., not significant."

[Supplementary Fig. 7f] in the revised manuscript,

"(f) Inhibition of soft agar colony formation by RT11-i compared to that with TMab4-i, as described in Fig. 4f. The pictures are representative of three independent experiments."

In the [Results] (pp. 11-12) of the revised manuscript,

"RT11-i exhibited significantly improved anti-proliferative activity against oncogenic Ras mutant tumor cells grown in monolayer culture conditions, but not against Ras^{WT} Colo320DM cells, when compared with that of TMab4-i (Supplementary Fig. 7e). However, the Ras-specific blocking effect was modest because TMab4-i itself also exerted anti-proliferative activity due to RGD10-mediated integrin blocking of anchorage-dependent growth^{39,40}. When assessed in anchorage-independent growth conditions on soft agar, RT11-i resulted in ~40–67% suppression of colony formation in oncogenic Ras mutant SW480 (KRas^{G12V}), LoVo (KRas^{G13D}), AsPC-1(KRas^{G12D}), PANC-1 (KRas^{G12D}), HT1080 (NRas^{Q61K}), and H1299 (NRas^{Q61K}) cells, but not in Ras^{WT} Colo320DM cells, when compared to TMab4-i (Fig. 4f and Supplementary Fig. 7f). These data demonstrate that RT11-i retains the anti-proliferative activity of RT11 by specifically blocking oncogenic Ras signaling in tumor cells."

Comment #2-6) In Figure 5, the dose of antibody in the blood stream is at the nM level, not a level seen to be active in vitro, so this is a large mismatch between in

vitro and in vivo.

Response: If we convert the unit of serum antibody concentration in Fig. 5a (PK profile) from $\mu\text{g/ml}$ into a molar concentration assuming that the total blood volume of nude mice of approximately 20 g (body weight) is ~ 2 ml [Kawamoto et al. (2011) BMC Cancer 11:359], a single I.V. injection of 20 mg/kg of RT11-i and TMab4-i into mice yielded an antibody concentration of ~ 1 μM in the blood stream for the initial 4 h period, and this subsequently drops to a nM levels in the hundreds for 2 days and then to nM levels in the tens for the rest periods. Many studies have also shown that the biodistribution of I.V.-injected antibody concentrations usually shows high tumor-to-blood ratios of approximately 2–4-fold [Batra SK et al. (2002) Current Opinion in Biotechnology, 13:603–608; Pastuskovas CV et al. (2012) Molecular Cancer Therapeutics, 11(3):752-762; Beckman RA et al. (2007) Cancer 109(2) 170-9; Press OW et al. (2001) Blood 98(8) 2535-43]. Thus, RT11-i concentration is expected to be much higher in tumor tissues than in the blood. Furthermore, in the tumor xenograft mouse experiments, RT11-i was intravenously dosed at 20 mg/kg every 2 days. Considering the terminal serum half-life (105.8 ± 2.3 h) of RT11-i, tumors are expected to be exposed to much higher concentrations of RT11-i than the concentration in the blood stream shown by a single injection. As shown in Fig. 6c and Supplementary Fig. 11a of the original manuscript, the IHC data of excised tumor tissues revealed the co-localization of RT11-i with Ras mutants at the inner plasma membrane of tumor cells, indicating that RT11-i is sufficiently bioavailable *in vivo*, inside the cytosol of tumor cells, after systemic intravenous administration.

We revised Fig. 5a by adding another y-axis to show molar concentrations of the antibody, as shown below.

	$T_{1/2\alpha}$ (h)	$T_{1/2\beta}$ (h)
TMab4-i (20 mg/kg)	3.9 ± 0.7	100.1 ± 3.1
RT11-i (20 mg/kg)	5.4 ± 0.3	105.8 ± 2.3

Regarding the difference between *in vivo* and *in vitro* efficacies, we observed that the anti-tumor activity of RT11/RT11-i was more dramatic in anchorage-independent soft-agar colony formation assays than in monolayer cell cultures (the *in vitro* data of the original manuscript), as shown in experiments performed for this revision. Since tumor growth in nude mice is correlated with anchorage-independent growth in soft agar, the *in vivo* anti-tumor efficacy of RT11-i in mouse models seems to be more significant than that observed *in vitro* under monolayer cell culture conditions.

In response to the reviewer's comment, we added the following sentence to the **[Results]** (p. 13) of the revised manuscript,

"The *in vivo* sensitivity of the tumors to RT11-i was in the following order: HT1080 (~70% TGI) > SW480 (~56% TGI) > LoVo (~46% TGI), which roughly correlated with *in vitro* colony formation assay results (Fig. 4f), indicating their different dependency on oncogenic Ras signaling."

Comment #2-7) The dosing for efficacy in Figure 6 is done Q2D or Q5D, which is extremely frequent for an antibody therapy, and the doses are extremely high. Yet, the efficacy is very very modest. For such a target, the efficacy of blocking K-Ras mutants should be strong tumor regression.

Response: In the original manuscript, there were typos regarding the description of

tumor-growth inhibition (TGI) efficacy of RT11-i assessed in the tumor xenograft mouse models. RT11-i inhibited the growth of human colorectal SW480 and LoVo tumors with a TGI of approximately 56% (not 22% in the original manuscript) and 46% (Fig. 6a), as well as a 42% (not 12% in the original manuscript) and 43% greater reduction in tumor weight (Supplementary Fig. 10b), respectively, compared to those after treatment with TMab4-i. In this revision, we assessed further the *in vivo* anti-tumor efficacy of RT11-i in nude mice bearing human soft tissue sarcoma HT1080 (NRas^{Q61K}) xenografts in the same manner (Q2D at 20 mpk) as SW480 and LoVo tumors. As shown in Fig. 6a (also in the Response to Reviewer #3 comment #3-10), RT11-i inhibited the growth of HT1080 tumor xenografts, showing approximately 70% more TGI and approximately 63% greater reduction in tumor weight, compared to those after treatment with TMab4-i.

However, as the reviewer commented, the anti-tumor efficacy of RT11-i is not extremely potent despite the frequent administration and high dose (Q2D at 20 mpk). This is probably due to the limited availability of RT11-i in the cytosolic space of tumor cells after systemic administration. To reach the cytosol and block oncogenic Ras signaling, RT11-i needs to be internalized via endocytosis and then released from endosomes, the efficiency of which is currently ~4.3% (Supplementary Table 2), limiting the availability of RT11-i in the targeted cytosol. Thus the endosome escape efficiency (currently ~4.3%) of RT11-i after receptor-mediated endocytosis requires improvement to achieve sufficient cytosolic localization for the greater anti-tumor efficacy.

Although therapeutic efficacies of RT11-i are not very high at this point, our results demonstrate the feasibility of developing therapeutic antibodies that directly target cytosolic oncogenic Ras mutants via systemic administration, like conventional therapeutic antibody regimens. In the future, we need to further engineer RT11-i to have increased potency for clinical applications. This point was discussed in the **[Discussion]** (p.17) of the revised manuscript, as shown in our response to your comment #2-10.

Comment #2-8) Since the antibody recognizes the GTP-bound state of Ras, the authors should check if the antibody affects GTP hydrolysis to K-Ras^{WT} and K-Ras^{G12D} *in vitro*.

Response: As the reviewer requested, we examined whether RT11 affects the intrinsic GTPase activity of KRas^{WT} and KRas^{G12D} *in vitro*, following the protocol described previously [Hunter JC et al. (2015) Mol Cancer Res 13(9):1325-1335]. In this assay, we incubated GTP-loaded KRas^{WT} and KRas^{G12D} proteins alone (for the intrinsic GTPase activity assay) or with RT11, using TMab4 as a control, and then monitored GTP hydrolysis for 20 min using EnzCheck phosphate assay kit (Thermo Scientific). As shown below, RT11 did not significantly affect the intrinsic GTP hydrolysis activity of both KRas^{WT} and KRas^{G12D}. However, in the positive control experiment, GTPase-activating protein of Ras (RasGAP) significantly stimulated GTP hydrolysis of GTP-bound KRas^{WT} and KRas^{G12D}. The values of intrinsic and RasGAP-stimulated GTP hydrolysis rate constants were comparable to those reported earlier [Hunter JC et al. (2015) Mol Cancer Res 13(9):1325-1335]. These data indicate that the inhibitory activity of RT11 against active Ras signaling is not due to an acceleration in GTP hydrolysis of active Ras, but rather due to inhibition of PPIs between active Ras and effector molecules.

The result was added to **[Supplementary Fig. 6]** of the revised manuscript,

Supplementary Figure 6. RT11 does not affect the intrinsic GTPase activity of KRas^{WT} and KRas^{G12D} proteins. (a,b) Kinetic profiles of GTP hydrolysis of GTP-loaded KRas^{WT} and KRas^{G12D} proteins in the absence (intrinsic GTPase activity) and presence of GTPase-activating RasGAP protein (GAP-stimulated GTPase activity) (a) and the indicated antibodies (b). In (a), GAP-stimulated GTP hydrolysis was accessed with the catalytic domain of RasGAP (residues 714–1047)⁵⁷. (c) Comparisons of GTP hydrolysis rate constants (k). GTP hydrolysis was determined by continuously measuring the release of phosphate using a purine nucleoside phosphorylase–based colorimetric assay⁵⁷. The concentration of phosphate released vs. time was plotted and the first-order rate constant was determined. In (c), error bars represent the mean \pm s.d. of three independent experiments. *** $P < 0.001$; n.s., not significant. The intrinsic and RasGAP-stimulated GTP hydrolysis rate constants of KRas^{WT} and KRas^{G12D} were comparable to those reported earlier⁵⁷.

We also described the above results in the **[Results]** (p. 10) of the revised manuscript:

"We examined whether RT11 affects the intrinsic GTPase activity of GTP-loaded KRas^{WT} and KRas^{G12D} *in vitro*. RT11 did not significantly affect the intrinsic GTP hydrolysis rate of both KRas^{WT} and KRas^{G12D}, similar to T Mab4 (Supplementary Fig. 6), indicating that the inhibitory activity of RT11 against active Ras signaling is not through an acceleration in GTP hydrolysis of active Ras."

In [Supplementary Methods] (pp.22-24 and p. 36) of the revised manuscript,

"Construction of recombinant protein expression plasmids

The human RasGAP (RASA1) gene (clone ID hMU002576) was purchased from Korea human gene bank of Medical Genomics Research Center (KRIBB, Korea). For mammalian expression, a DNA fragment encoding the catalytic domain of RasGAP (residues 714–1047) was subcloned in frame into the *AscI/ApaI* site of the pSecTag2A vector to add a 6× His tag to the N-terminus, generating pSecTag2A RasGAP(714–1047)."

"Protein expression and purification

To purify the catalytic domain of RasGAP (residues 714–1047), the pSecTag2A RasGAP(714-1047) plasmid was transiently transfected into 200 ml of HEK293F cell culture in Freestyle 293F media (Invitrogen) following the standard protocol, as previously described^{65,76}. After 5 d of culture, culture supernatant was harvested by centrifugation and filtration (0.22 μm, Polyethersulfone, Corning, CL S43118). The supernatant was incubated for 1 h with Ni-NTA resin (Clontech), the loaded beads were then washed with lysis buffer (20 mM Tris, pH 8.0, 300 mM NaCl, 5 mM imidazole) and the proteins were eluted with elution buffer (20 mM Tris, pH 7.4, 2 mM MgCl₂, 250 mM imidazole). Eluted proteins were dialyzed in buffer (50 mM Tris, pH 8.0, 1 mM DTT) and concentrated to 1 mg/ml using a Spin-X 20 ml centrifugal concentrator (Corning)."

" GTPase assay of Ras proteins

GST-fused KRas (WT and G12D mutant) was loaded with GTP (Sigma) following the protocol for the preparation of GppNHp-loaded Ras proteins, as described earlier. GTPase activity of Ras proteins was measured using EnzCheck phosphate assay kit (Thermo Scientific, E6646) following the protocol described previously⁵⁷. Briefly, assay buffer only (intrinsic GTPase activity), indicated antibody (15 μM final concentration), or RasGAP (15 μM final concentration) was added to reaction mixture (30 μM GTP-loaded KRas proteins (WT or G12D), 200 μM 2-amino-6-mercapto-7-methylpurine riboside (MESG), and 50 U/ml purine nucleoside phosphorylase (PNP) in assay buffer (50 mM Tris-HCl, pH 8.0, 1 mM DTT)) in 96-well half-area microplates (Corning). GTP hydrolysis was initiated by the

addition of MgCl₂ (10 mM final concentration). The absorbance at 360 nm was measured at 20 °C for 20 min at every 10 s interval on a MULTISKAN GO plate reader (Thermo Scientific). The phosphate concentration ([Pi]_t) at each time point was determined by comparison with a phosphate standard curve and plotted against time. The hydrolysis rate constant (*k*) was determined by fitting the data to a single-phase, exponential non-linear regression curve with the equation $[Pi]_t = [Pi]_0 + ([Pi]_{final} - [Pi]_0) (1 - \exp(-kt))$ in GraphPad Prism (GraphPad software, Inc.)⁵⁷."

Comment #2-9) The authors state that the affinity for H-Ras to the RAF-RBD is 160 nM. The authors should include citations on more recent papers on Raf and PI3K binding affinity particularly:

- a. Kiel C, Filchtinski D, Spoerner M, Schreiber G, Kalbitzer HR, Herrmann C. Improved Binding of Raf to Ras-GDP Is Correlated with Biological Activity. *The Journal of Biological Chemistry*. 2009;284(46): 31893-31902. doi:10.1074/jbc.M109.031153.
- b. Hunter, J. C. et al. Biochemical and Structural Analysis of Common Cancer-Associated KRAS Mutations. *Mol. Cancer Res*. 13, 1325–1335 (2015).

Response: As the reviewer requested, we carefully searched and cited recent literatures regarding the affinity of Ras-Raf, Ras-PI3K, and Ras-RalGDS. Specifically, we included the two papers mentioned by the reviewer for the interaction of Ras-Raf, one paper for Ras-PI3K [Fritsch R et al. (2013) *Cell* 153: 1050–1063], and one paper for Ras-RalGDS [Linnemann T et al. (2002) *The Journal of biological chemistry* 277, 7831-7837.].

In the **[Results]** (p. 6) of the revised manuscript:

"When measured by surface plasmon resonance (SPR), the *K_D* values ranged from 4 to 17 nM (Supplementary Table 1), which suggests a much stronger affinity than that of active Ras for its effector proteins, such as approximately 56–160 nM for Raf²⁶⁻²⁸, 1 μM for RalGDS²⁹, and 2.7 μM for PI3K^{30,31}. RT11 efficiently competed with the RBDs of cRaf (cRaf_{RBD}) and

RalGDS (RalGDS_{RBD}), as well as PI3K α (p110 α /p85 α), for binding to KRas^{G12D}·GppNHp (Fig. 1d), suggesting that RT11 binds to the PPI interfaces between active Ras and effector proteins."

Comment #2-10) The authors show that in HeLa cells approximately 1:10 of extracellular concentration of RT11 is retained in the cytosol (Supplementary Table 2). The concentration of Ras in HeLa cells is 0.4-1.6 μ M (Fujioka A et al. Journal of Biological Chemistry 281, 8917–8926 (2006)). It is a challenge to inhibit this large quantity of Ras with the current antibody, which has still very poor permeability across the membrane. The authors might want to discuss how the TMabs internalization can be improved through future rounds of antibody engineering.

Response: As the reviewer commented, improved cytosolic release of RT11/RT11-i iMabs after cellular endocytosis will lead to higher cytosolic concentrations, which would trigger more efficient blocking of oncogenic Ras mutants, and thereby result in increased anti-tumor potency. Judging from our previous studies with TMab4 cytotransmab [Kim JS et al. (2016) J Control Release 235:165-175], the most limiting factor of cytosolic localization is the poor endosome escape efficiency after cellular internalization via receptor-mediated endocytosis. The endosome escape efficiency (%) of TMab4, estimated as a percentage (%) by dividing the cytosolic-released amount with the total internalized amount [Kim JS et al. (2015) Biochemical and biophysical research communications 467:771-777], was approximately 4.3% at the initial extracellular concentration of 1 μ M. Since RT11/RT11-i iMabs have the same cytosol-penetrating VL as that of TMab4, they also showed a similar endosomal escape efficiency (Supplementary Table 2). Of note, we recently generated a TMab4 variant, called TMab4-WYW, with ~3-fold improved endosomal escape efficiency (~13%) compared to that of TMab4 by engineering the VL [Kim JS et al. (2016) J Control Release 235:165-175]. We are now investigating whether the improved VL of TMab4-WYW can be assembled with the Ras·GTP specific-binding VH of RT11/RT11-i to generate more potent Ras-targeting iMabs.

To address the reviewer request, we added the following to the **[Discussion]**

(p. 17) in the revised manuscript:

"Nonetheless, the *in vivo* anti-tumor activity of RT11-i is not very potent as a single agent, and requires frequent dosing at relatively high concentrations for significant suppression of tumor growth (Fig. 6). Thus RT11-i should be further engineered to improve potency for practical clinical studies. For this purpose, the endosome escape efficiency (currently ~4.3%) of RT11-i after receptor-mediated endocytosis requires improvement to achieve sufficient cytosolic localization to address relatively high cellular concentrations (0.4–1.6 μM) of Ras⁴⁵. Such RT11-i derivative can be generated by incorporating the recently engineered cytosol-penetrating TMab4 VL variant with ~3-fold improved endosomal escape efficiency (~13%)²⁴. Furthermore, affinity maturation of RT11-i against the active form of oncogenic Ras mutants (currently 4–17 nM) will be necessary to increase the efficiency of blocking Ras·GTP–effector PPIs."

Responses to specific comments from reviewer #3

General comments: The manuscript from Shin and colleagues describes the generation of an antibody (RT11) targeting RAS proteins with a quite beautiful specificity for RAS mutant tumours. The authors characterize the effect of such antibody in *in vivo* and *in vitro* assays, showing decreased tumour proliferation of cells in both experimental settings. They also showed how this antibody impairs downstream signalling of the two major RAS effector pathways: ERK and PI3K. Although the results shown are potentially interesting for the field, there are a few issues that are not clearly addressed in the manuscript at the moment and would be important to address:

Comment #3-1) The authors claim through the text that their antibody is specific for KRas. However in figure 1c they show it also binds to HRas and NRas and they showed in figure 3a and supplementary Fig. 4a that RT11 inhibits proliferation of NRas mutated cell lines. So it is not specific for KRas. This should be removed from the text.

Response: Thank you for your comment. As you mentioned, RT11/RT11-i exhibits broad binding specificity to the active forms of wild-type KRas, NRas and HRas and their respective oncogenic derivatives. Further, additional experiments revealed that RT11/RT11-i iMabs suppress the growth of oncogenic NRas mutant tumor cells as well as KRas mutant tumor cells *in vitro* and *in vivo* (please see the revised Fig. 3a, Fig. 4f, Fig. 6, and Supplementary Fig. 5), demonstrating that RT11/RT11-i exhibits broad specificity to oncogenic Ras mutants. Accordingly, we changed the description of RT11/RT11-i from “active KRas·GTP specific” to “active Ras·GTP specific” in the context of the antigen specificity and the anti-tumor activity in the appropriate sentences throughout the revised manuscript. Please refer to the revised manuscript, in which the changes were highlighted in red color. We apologize for not showing all of the revised parts in this response letter due to space limitations.

Comment #3-2) Since the antibody was generated against Ras-GTP and it can recognize HRas, NRas and KRas, authors should check how specific the antibody is and if it still can recognize and bind other Ras-GTP superfamily members such as MRas, RRas, or even Rac.

Response: As we demonstrated in the original manuscript (Fig. 1c and Supplementary Fig. 7c), RT11/RT11-i specifically recognizes the active forms of wild-type KRas, NRas, and HRas and their oncogenic mutants with representative oncogenic mutations at residue 12, 13, or 61. This broad specificity of RT11/RT11-i to active Ras is accounted for by the binding epitopes on the PPI interfaces, namely the switch I and II regions, of active Ras with effector proteins (Fig 1e and Supplementary Fig. 3b). The four Ras proteins (KRas4A, KRas4B, HRas, and NRas) are highly homologous throughout the G domain (amino acids 1–165). The first 85 amino acids are identical in all four proteins. Thus, they share identical PPI interface residues in the switch I/II regions, including the putative binding epitopes (D33, P34, T35, I36, E37, D38, Y40, E63) of RT11 (Fig. 1e; please refer to the newly added figure in Supplementary Fig. 4c in the revised manuscript). When measuring the affinity by surface plasmon resonance (SPR), the K_D values ranged from 4 to 17 nM for the active forms of the wild-type and oncogenic Ras mutants of KRas, NRas, and HRas (Supplementary Table 1). Though the K_D values varied slightly depending on the isotype and type of mutation, it is difficult to discern differences in terms of the specificity of RT11/RT11-i to Ras isotypes and oncogenic mutations at this point. Thus, as stated in the original manuscript, we concluded that RT11/RT11-i exhibited broad specificity to the active forms of Ras proteins, KRas, HRas, and NRas, regardless of isotype and oncogenic mutations. In response to the reviewer's comment, we added a figure showing sequence comparisons for the switch I and II regions of KRas with those of other Ras superfamily proteins such as HRas, NRas, MRas, RRas, and Rac1, in **[Supplementary Fig. 4c]** of the revised manuscript.

The reviewer also requested the binding specificity of RT11/RT11-i for other Ras superfamily small GTPases such as MRas, RRas, or even Rac. MRas and RRas are Ras family members, whereas Rac protein belongs to Rho family

members [Karnoub AE et al. (2008) Nat. Rev. Mol. Cell. Biol. 9:517-31]. To address the reviewer's comment, we determined binding specificity of RT11/RT11-i iMabs for the active and inactive forms of MRas, RRas, and Rac1. We prepared the three human proteins by bacterial expression and performed ELISA with the GppNHp- and GDP-bound forms.

As shown below, RT11/RT11-i exhibited ~50% binding to the GppNHp-bound active forms, but did not bind to the GDP-bound inactive forms, of the Ras family member proteins, MRas^{WT} and RRas^{WT}, when compared to that for active KRas^{G12D}. These two proteins share almost identical primary structures in the switch I and II regions with those of Ras proteins. However, MRas exhibits slightly distinct tertiary conformations in the switch I and II regions compared to those of Ras proteins [Matsumoto K et al. (2011) J. Biol. Chem. 286(17):15403-12]. The crystal structure of GppNHp-bound RRas has not yet been determined, but we expect that the tertiary structure of RRas would be similar to that of MRas rather than that of KRas based on the primary sequence homology (60.9% and 51.8% sequence identity with MRas and KRas, respectively). Thus, the ~50% binding activities of RT11/RT11-i for the active forms of MRas^{WT} and RRas^{WT} compared with those of active KRas could be accounted for by the almost identical residues in the switch I and II regions of the Ras family members, but the slightly distinct tertiary conformations, compared to those of Ras proteins. RT11/RT11-i did not cross-react with either active or inactive form of Rac1, which belongs to the Rho family of GTPases and has much different primary and tertiary structures than those of Ras proteins (30.3% sequence identity with KRas). The above results further supported the notion that RT11/RT11-i specifically recognizes the conformationally distinct switch I and II regions of active Ras proteins from the inactive forms.

We added the above results to **[Supplementary Fig. 4]** of the revised manuscript,

"Supplementary Figure 4. RT11/RT11-i binds weakly to the GppNHp-bound active forms of Ras superfamily members MRas and RRas, but does not cross-react with either active or inactive form of the Rho family member Rac1. (a) Reducing SDS-PAGE analyses of bacterially expressed and purified MRas^{WT}, RRas^{WT}, and Rac1^{WT}. **(b)** Binding activity of the indicated antibodies to the GppNHp-bound active forms or GDP-bound inactive forms of MRas^{WT}, RRas^{WT}, and Rac1^{WT}, compared to that of active KRas^{G12D}. ELISA plates were coated with 5 µg/ml of antibodies and then incubated with the antigen at

100 nM. Error bars represent the mean \pm s.d. ($n = 3$). (c) Comparison of KRas residues in the switch I and II regions with those of HRas and NRas as well as other Ras superfamily GTPases such as MRas, RRas, and Rac1. The residues identified as putative binding epitopes of RT11 on KRas^{G12D} by alanine scanning mutagenesis are highlighted as red circles. The residue numbering is based on the sequence of KRas. (d) Comparison of the crystal structures of GppNHp-bound active KRas (PDB ID: 3GFT), HRas (PDB ID: 1CTQ)⁵², MRas (PDB ID: 1X1S)⁵³, and Rac1 (PDB ID: 5FI0)⁵⁴, highlighting the putative binding epitopes of RT11 on KRas (right panel). The switch I and II regions of KRas superimpose well with HRas showing root-mean-square-deviations of 0.45 Å, but poorly with MRas and Rac1 showing root-mean-square-deviations of 2.06 and 2.47 Å, respectively. The crystal structure of GppNHp-bound RRas has not yet been determined, but it is expected to be similar to that of MRas rather than to that of KRas based on primary sequence homology (60.9% and 51.8% sequence identity with MRas and KRas, respectively)^{55,56}. The images were generated using the PyMol program (Schrödinger)."

We described the above results in **[Results]** (pp. 6-7) of the revised manuscript, "When assessed with other Ras family members³², RT11 exhibited ~50% binding activity for GppNHp-bound active forms of MRas^{WT} and RRas^{WT}, but not for the GDP-bound inactive ones, when compared to active KRas^{G12D} (Supplementary Fig. 4a,b). This can be explained by the virtually identical residues in the switch I and II regions of the Ras family members, but the slightly distinct tertiary conformations, compared to those of Ras proteins (Supplementary Fig. 4c,d)³³. However, RT11 did not cross-react with both active and inactive forms of Rac1^{WT}, belonging to the Rho family of GTPases³², which has much different primary and tertiary structures from those of Ras proteins (Supplementary Fig. 4)."

In **[Supplementary Methods]** (pp. 22-24) of the revised manuscript,

"Construction of recombinant protein expression plasmids

Human RRas (clone ID hMU001042) and Rac1 (clone ID KU000510) genes were purchased from Korea human gene bank of Medical Genomics Research Center (KRIBB, Korea). The human MRas gene was prepared by DNA synthesis (Bioneer Inc.). For bacterial expression, DNA fragments encoding MRas (residues 1–208), RRas (residues 1–218), and Rac1 (residues

1–177) were individually subcloned in frame into the *NheI/BamHI* sites of the pET23 vector to add a 6× His tag to the N-terminus, generating pET23-His-MRas, pET23-His-RRas, and pET23-His-Rac1, respectively."

"Protein expression and purification

To purify the His-tagged proteins (His-KRas^{G12D} (residues 1–169), His-KRas^{G12D}-based Ala mutants, His-MRas, His-RRas, and His-Rac1), the pET23-His-based plasmids were individually transformed into *E coli* BL21 (DE3)plysE cells.

The purified His-MRas, His-RRas, and His-Rac1 were used for direct ELISA."

Comment #3-3) In figure 2b the authors showed co-localization of RT11 with mutant KRas at the inner plasma membrane. Although localization of their antibody seems different from their control, images are very small and it is very difficult to determine proper localization of antibodies. Staining looks all over the cytoplasm and not inner plasma membrane as authors claim. Images with a higher magnification are required and/or images with enlarged areas where results can be easily observed. Also arrows indicating areas with the results they want to show will help readers to quickly spot differences.

Response: In response to the reviewer's comment, we repeated the experiments shown in Fig. 2b and replaced the original images with high-quality ones with enlarged areas and appended arrows to clearly demonstrate the co-localization of RT11 with activated Ras around the inner plasma membrane. Although Ras in the cytosolic region was stained partially, RT11, but not TMab4, predominantly co-localized with Ras at the plasma membrane. Further we evaluated the localization of RT11 in Ras^{WT}-harboring colorectal HT29 cells to compare the localization of RT11 between oncogenic Ras mutant and Ras^{WT}-harboring tumor cells, as shown in Fig. 2b.

[Fig. 2b] and its legend in the revised manuscript:

"(b) Cellular internalization and co-localization of RT11 (green), but not TMAb4 (green), with the inner plasma membrane-anchored active Ras (red) in mCherry-KRas^{G12V}-transformed NIH3T3 and KRas^{G12V}-harboring SW480 cells, analyzed by confocal microscopy. The Ras^{WT}-harboring HT29 cells were also analyzed as a control. The areas in the white boxes are shown at a higher magnification for better visualization. The arrow indicates the co-localization of RT11 with activated Ras. Scale bar, 5 μ m."

In the **[Results]** (pp. 7-8) of the revised manuscript,

"Furthermore, when incubated with mCherry-KRas^{G12V}-transformed mouse fibroblast NIH3T3 and KRas^{G12V}-harboring human colorectal SW480 cells under normal cell culture conditions, internalized RT11, but not TMAb4, co-localized with KRas^{G12V} at the inner plasma membrane (Fig. 2b), where active KRas^{G12V}·GTP is anchored^{1,2}. However, in Ras^{WT}-harboring human colorectal HT29 cells, RT11 predominantly localized throughout the cytosol without enrichment at the plasma membrane (Fig. 2b), similar to TMAb4. This could be ascribed to the predominant presence of Ras^{WT} in the inactive form in resting cells^{34,35}."

Comment #3-4) Soft agar colony formation assays with other Ras mutations should

be shown to determine specificity of oncogenic KRas recognition.

Response: As the reviewer requested, we performed further soft agar colony formation assays for tumor cells with various oncogenic Ras mutations, such as LoVo (KRas^{G13D}), HT1080 (NRas^{Q61K}), and H1299 (NRas^{Q61K}) cells, in addition to SW480 (KRas^{G12V}) and PANC-1 (KRas^{G12D}) cells shown in supplementary Fig. 4 of the original manuscript. Ras^{WT}-harboring K562 cells were further used as a negative control in addition to Colo320DM (Ras^{WT}) cells. As shown below, RT11 treatment resulted in an approximate 44–64% inhibition in colony formation for diverse oncogenic Ras mutant tumor cells, but not for Ras^{WT} cells, when compared to that with T Mab4. These results clearly demonstrate that RT11 inhibits the growth of tumor cells harboring oncogenic Ras mutants, but not those with Ras^{WT}.

This data was added to **[Supplementary Fig. 5c]** of the revised manuscript:

"(c) Inhibition of soft agar colony formation of tumor cells by RT11. Anchorage-independent cell growth was examined by soft agar colony formation assays using the oncogenic KRas mutant, **NRas mutant**, and Ras^{WT} cells, treated with RT11 (2 μM) and T Mab4 (2 μM) every 72 h for 2–3 weeks. Following treatment, the number of colonies (**diameter > 200 μm**) was counted after BCIP/NBT staining, as shown in the pictures of the representative soft agar plates (right)."

In the **[Results]** (p. 9) of the revised manuscript,

"Oncogenic Ras mutations are known to drive anchorage-independent tumor growth, another important hallmark of cellular transformation^{21,36}. Based on soft agar colony formation assays, RT11 treatment resulted in ~44–64% inhibition of anchorage-independent proliferation for oncogenic KRas mutant cells (SW480, LoVo, and PANC-1) and NRas mutant cells (HT1080 and H1229), but not in Ras^{WT} Colo320DM and K562 cells, compared to that after TMab4 treatment (Supplementary Fig. 5c). This result demonstrates that RT11 suppresses the tumorigenic activity of oncogenic Ras mutants. Of note, the effect on tumor growth after blocking oncogenic Ras with RT11 was more evident in anchorage-independent growth conditions (soft agar) than that in monolayer culture conditions (Supplementary Fig. 5a,c), which is line with previous observations with siRNA-mediated knockdown of oncogenic Ras^{37,38}."

Comment #3-5) In figure 3A authors show decreased proliferation of cells treated with their RT11 antibody. In order to evaluate the proliferation effect of the antibody with other conventional therapies it would be nice to compare results with Akt and/or ERK inhibitors.

Response: As the reviewer requested, we examined the effect of pharmacological inhibitors, sorafenib (Raf kinase inhibitor) and LY294002 (PI3K-Akt inhibitor), on the *in vitro* proliferation of tumor cells and compared these results to those with RT11 antibody. As shown below, the pharmacological inhibitors had anti-proliferative effects on oncogenic Ras mutant SW480, LoVo and AsPC-1 cells, as well as Ras^{WT} Colo320DM cells. However, unlike the pharmacological inhibitors, RT11 did not show significant cytotoxicity to Ras^{WT} cells, specifically colorectal Colo320DM and HT29 cells, breast MCF-7 cells, and leukemic K562 cells, as well as non-transformed NIH3T3 cells (Supplementary Fig. 5a). These results suggest that RT11 specifically inhibits the proliferation of oncogenic Ras-driven cells, but not Ras^{WT} Colo320DM cells.

[Supplementary Fig. 5b] and its legend in the revised manuscript,

"(a, b) Cellular proliferation assay in indicated cells, treated twice at 0 and 72 h with the indicated concentration of antibody (RT11 or TMab4) (a) or the pharmacological inhibitor (Raf kinase inhibitor sorafenib or PI3K-Akt inhibitor LY294002) (b) for 6 d. Error bars represent the mean \pm s.d. ($n = 3$). ** $P < 0.01$, *** $P < 0.001$."

In the **[Results]** (pp. 8-9) of the revised manuscript:

" The anti-proliferative activity of RT11 was much weaker than and comparable to that of the pharmacological inhibitor sorafenib (Raf kinase inhibitor) and LY294002 (PI3K-Akt inhibitor), respectively, when compared at the equivalent molar concentrations (Supplementary Fig. 5b). However, unlike the pharmacological inhibitors, RT11 did not show significant cytotoxicity to Ras^{WT} cells, specifically colorectal Colo320DM and HT29 cells, breast MCF-7 cells, and leukemic K562 cells, as well as non-transformed NIH3T3 cells (Fig. 3a and Supplementary Fig. 5a), indicating that direct Ras blocking by RT11 results in minimal toxicity to Ras^{WT} cells probably due to their minimal dependence on the Ras-driven signaling for proliferation^{1,2,35}."

Comment #3-6) In figure 3e and f the effect of the antibody on Ras downstream pathway inhibition seems to have a strong variation. Cells have been treated with different growth factors and one wonders if the differences are because of the use of

FBS or EGF. On NIH3T3-HAKRasG12V cells decrease in p-ERK and p-Akt seems minimal. Downstream targets of pAkt specially need to be explored in order to determine downregulation of the pathway.

Response: Although EGF is generally used activate ERK1/2 and Akt in human tumor cells including KRas^{G12V}-harboring SW480 cells, it is known to exert minimal effects on wild-type or oncogenic KRas mutant-overexpressing mouse fibroblast NIH3T3 cells, due to the absence of EGFR [Evdokimova V et al. (2006) Mol. Cell. Biol. 26(1):277-292]. Thus, based on the literature [Bosch M et al. (1998) J. Biol. Chem. 273(34):22145-150], we used 10% FBS to stimulate active Ras downstream effector signaling in KRas^{G12V}-transformed NIH3T3 cells.

Regarding the reviewer's concern about the minimal inhibitory effects of RT11 on downstream signaling in Fig. 3e,f, as we responded to comment #2-4 (from reviewer 2), quantification of band intensities of the Western blotting data revealed that RT11 attenuated the activation of MEK1/2, ERK1/2, and Akt in a dose-dependent manner. The quantification data of the original results is shown in the response to reviewer comment #2-4.

To address the reviewer's concern, we repeated the experiments shown in Fig. 3e,f with higher RT11 antibody concentrations (2 and 10 μ M) than those (1 and 5 μ M) used in the original manuscript to clearly show the dose-dependent inhibition on downstream Ras signaling by RT11. As the reviewer requested, we also monitored the activation of p70S6K, a downstream kinase of the PI3K-Akt-mTOR pathway [Chung J et al. (1994) Nature 370:71-75]. As shown below, RT11 exhibited dose-dependent inhibition of serum- and EGF-stimulated MEK1/2, ERK1/2, Akt, and p70S6K phosphorylation in HA-KRas^{G12V}-transformed NIH3T3 cells (Fig. 3e) and SW480 cells (Fig. 3f), respectively. These results confirm that RT11 attenuated the downstream signaling mediated by Ras·GTP-effector PPIs, such as Raf-MEK1/2-ERK1/2 and PI3K-Akt-mTOR pathways in a dose-dependent manner.

The new data replaced the original data in [Fig. 3e,f] of the revised manuscript.

"(e,f) Inhibitory effect of RT11 on the downstream signaling of KRas-effector PPIs in HA-KRas^{G12V}-transformed NIH3T3 cells (e) and SW480 cells (f), analyzed by western blotting. The cells were serum-starved for 6 h before treatment with antibody, Raf kinase inhibitor sorafenib, or PI3K-Akt inhibitor LY294002 for 6 h in serum-free growth medium. Cells were washed and then stimulated with 10% FBS (e) and EGF (50 ng/ml in serum free-media) (f) for 10 min before cell lysis. The number below the panel indicates relative value of band intensity of phosphorylated proteins compared to that in the PBS-treated control after normalization to the band intensity of respective total protein for each sample. * $P < 0.05$, ** $P < 0.01$, *** $P < 0.001$ vs. PBS-treated control cells. In (b-f), images are representative of at least two independent experiments."

In the [Results] (p. 10) of the revised manuscript,

" We next investigated the effect of RT11 on downstream signaling mediated by Ras·GTP-effector PPIs such as the Raf-MEK1/2-ERK1/2 and PI3K-Akt-mTOR pathways^{1,2}. RT11 exhibited dose-dependent inhibition of downstream signaling mediated by PPIs of Ras-Raf (MEK1/2 and ERK1/2) and Ras-PI3K (Akt and p70S6K) in serum-stimulated HA-KRas^{G12V}-transformed NIH3T3 cells (Fig. 3e) and epidermal growth factor (EGF)-stimulated SW480 cells (Fig. 3f). The Raf inhibitor sorafenib and PI3K-Akt inhibitor LY294002 attenuated only targeted signaling in SW480 cells (Fig. 3f)."

Comment #3-7) On figure 4d images are very low magnification to properly see inner membrane localization of the antibodies. RT11-i and TMab4-i seems to be localized in the same places. Localization of the antibodies on Ras WT cells nice would be good.

Response: In response to the reviewer's comment, we repeated the experiments shown in Fig. 4d and replaced the original images with high-quality ones with enlarged areas and appended arrows to clearly demonstrate the co-localization of RT11-i with activated Ras around the inner plasma membrane.

As the reviewer also recommended, we also evaluated the localization of RT11-i in Ras^{WT}-harboring colorectal HT29 cells. Unlike KRas^{G12V}-harboring SW480 cells, the co-localization of RT11-i with active Ras at the inner plasma membrane was negligible in Ras^{WT} HT29 cells. Instead RT11-i was predominantly detected in the cytosolic space without co-localization with Ras, similar to TMab4-i, most likely because Ras^{WT} predominantly exists in the inactive form in resting state cells [Hayes TK and Der CJ (2013) *Cancer Discov* 3(1):24-26; Hirasawa K et al. (2002) *Cancer Research* 62:1696-1701]. These results further confirm that RT1-i specifically recognizes the active form of Ras at the inner plasma membrane after cellular internalization and subsequent cytosolic localization.

[Fig. 4d] and its legend in the revised manuscript:

"(d) Cellular internalization and co-localization of RT11-i, but not T Mab4-i, with the inner plasma membrane-anchored active KRas·GTP in KRas^{G12V}-harboring SW480 cells. The Ras^{WT}-harboring HT29 cells were also analyzed as a control. The areas in the white boxes are shown at increased magnification for better visualization. The arrow indicates the co-localization of RT11-i with activated Ras. Nuclei were counterstained with Hoechst 33342 (blue). Scale bar, 5 μ m."

Comment #3-8) On figure 4f the difference on proliferation between RT-11-i and T Mab4-i is very small, especially if we compared the difference in proliferation of both antibodies before targeting them with the integrin sequence. Authors should show if proliferation decrease is increased with higher antibody concentrations as done in figure 3a.

Response: As the reviewer commented (also the same comment #2-5 from reviewer #2), the anti-proliferative activity of RT11-i at 1 μ M is not very dramatic, when compared to that of T Mab4-i. As we responded to comment #2-5, this is mainly due to the anti-proliferative activity of the control T Mab4-i, which also inhibits cell growth by the RGD10-mediated integrin-blocking effect of inhibiting integrin-mediated cell adhesion, particularly in cells overexpressing integrin α v β 5, such as SW480 cells

(Fig. 4b), under monolayer culture conditions [Desgrosellier JS et al. (2010) Nature reviews Cancer 10:9-22; Goodman SL and Picard M (2012) Trends in Pharmacological Sciences 33(7):405-412; Howe A et al. (1999) Current Opinion in Cell Biology, 10:220-231]. Indeed, when tumor cells were treated with TMab4-i at the higher concentration of 2 μ M, cellular growth was significantly inhibited by RGD10-mediated integrin-blocking activity triggering detachment of the cells from the culture plate, as shown below.

To address the reviewer's concern, the anti-tumor activities of RT11-i were further assessed and compared to those of TMab4-i using soft-agar colony forming assay (anchorage-independent growth conditions), in which we expected a more obvious oncogenic Ras-blocking effect, while minimizing the integrin-blocking effect. This is due to the anchorage-independent growth conditions that are not present in the monolayer culture conditions [Fujita-Sato S et al. (2015) Cancer Res. 75(14): 2851–2862; Shi XH et al. (2009) Cancer Gene Therapy 16:227-236; Fujita M et al. (1999) Melanoma Res. 9(3):279-291]. We tested diverse oncogenic Ras mutant cells (SW480 (KRas^{G12V}), LoVo (KRas^{G13D}), AsPC-1(KRas^{G12D}), PANC-1 (KRas^{G12D}), HT1080 (NRas^{Q61K}), and H1299 (NRas^{Q61K}) cells, and used Colo320DM (Ras^{WT}) cells as a control. As shown below, RT11-i exhibited an approximate 40–67% suppression of colony formation in oncogenic Ras tumor cells, but not in Ras^{WT} Colo320DM cells, when compared to the suppressive effect of TMab4-i. These data clearly demonstrate that RT11-i suppresses the proliferation of cancer cells by

blocking oncogenic Ras mutants.

The data obtained by soft-agar colony forming assays were added in Fig. 4f and the original Fig. 4f replaced Supplementary Fig. 7e, which was described appropriately in the revised manuscript.

[Fig. 4f] and its legend of the revised manuscript,

"(f) Inhibition of tumor cell soft agar colony formation by RT11-i compared to that with TTab4-i. Following treatment of cells with PBS, TTab4-i (2 μM), or RT11-i (2 μM) every 72 h for 2–3 weeks, the number of colonies (diameter >200 μm) was counted by BCIP/NBT staining, as shown in the pictures of representative soft agar plates (Supplementary Fig. 7f). The results are presented as percentages compared to the PBS-treated control. Error bars represent the mean ± s.d. ($n = 3$). * $P < 0.05$, ** $P < 0.01$; n.s., not significant."

[Supplementary Fig. 7f] and its legend in the revised manuscript,

"(f) Inhibition of soft agar colony formation by RT11-i compared to that with TMab4-i, as described in Fig. 4f. The pictures are representative of three independent experiments."

In the **[Results]** (pp. 11-12) of the revised manuscript,

"RT11-i exhibited significantly improved anti-proliferative activity against oncogenic Ras mutant tumor cells grown in monolayer culture conditions, but not against Ras^{WT} Colo320DM cells, when compared with that of TMab4-i (Supplementary Fig. 7e). However, the Ras-specific blocking effect was modest because TMab4-i itself also exerted anti-proliferative activity due to RGD10-mediated integrin blocking of anchorage-dependent growth^{39,40}. When assessed in anchorage-independent growth conditions on soft agar, RT11-i resulted in ~40–67% suppression of colony formation in oncogenic Ras mutant SW480 (KRas^{G12V}), LoVo (KRas^{G13D}), AsPC-1(KRas^{G12D}), PANC-1 (KRas^{G12D}), HT1080 (NRas^{Q61K}), and H1299 (NRas^{Q61K}) cells, but not in Ras^{WT} Colo320DM cells, when compared to TMab4-i (Fig. 4f and Supplementary Fig. 7f). These data demonstrate that RT11-i retains the anti-proliferative activity of RT11 by specifically blocking oncogenic Ras signaling in tumor cells."

Comment #3-9) On figure 5b it would be good to see how the RT11 antibody localizes compared to the RT11-i.

Response: In response to the reviewer's comment, the tumor-targeting ability of RT11-i was compared to that of RT11 in athymic nude mice bearing integrin $\alpha\beta 5$ -expressing SW480 xenografts, the same mouse model used in Fig. 5b of the original manuscript. DyLight 755-labeled antibodies (20 $\mu\text{g}/\text{mouse}$) were intravenously injected into the mice, and the whole body fluorescence was examined at 6, 12, and 24 h post-injection. Since tumor localization of RT11-i/TMab4-i was peaked at 24 h post-injection (Fig. 5b of the original manuscript), like other tumor-targeting antibodies (Kwon JH et al. (2013) *Blood* 43 1523-1530; Press OW et al. (2001) *Blood* 98(8) 2535-43; Cheng J et al. (2007) *Laryngoscope* 117(6) 1013-8), we compared the tissue distribution of RT11-i to that of RT11 in earlier time points such as 6, 12 h, and 24 h.

As shown below, RT11 without integrin-targeting RGD10 fusion did not show any increased distribution in the tumors compared to that in normal tissues during 24 h circulation. However, RT11-i showed preferential tumor tissue accumulation at 24 h post-injection showing the tumor-to-normal tissue ratio of approximately 2.4:1. *Ex vivo* analysis of fluorescence intensities for isolated tumors and normal organs at 24 h post-injection also showed ~3.4-fold higher tumor tissue accumulation of RT11-i than that of RT11. The above results demonstrate that RGD10-fused RT11-i can target tumor tissues overexpressing integrin $\alpha\beta 5$ *in vivo*.

The above data were added in **[Supplementary Fig. 9a,b]** with an appropriate explanation in the revised manuscript.

"(a,b) Comparison of biodistribution between RT11 and RT11-i, evaluated by intravenously injecting Dylight755-labeled antibodies (20 $\mu\text{g}/\text{mouse}$) into SW480 xenograft tumor-bearing mice. In (a), representative whole body fluorescence images, which were acquired at the indicated times post-injection. Fluorescence intensities in the tumor tissue (T), as indicated by arrows, and normal tissues (N) were quantified. In (b), *ex vivo* analysis of fluorescence intensities of dissected tumors and normal organs, which were acquired at 24 h post-injection. The right panel shows quantified fluorescence intensities of each tissue or organ. In (a,b), error bars represent the mean \pm s.d. ($n = 6$ per group)."

In the **[Results]** (p. 12) of the revised manuscript,

"We next determined tissue distribution by intravenous dosing of DyLight755-labeled antibodies into nude mice bearing integrin $\alpha\text{v}\beta\text{5}$ -expressing SW480 xenograft tumors. RT11 without RGD10 fusion did not exhibit any increased distribution in the tumors, compared to that in normal tissues, during 24 h circulation (Supplementary Fig. 9a,b). In contrast, RGD10-fused RT11-i and TMab4-i displayed preferential accumulation in the tumors with a peak at 24 h post-injection, when compared to that in the normal tissues (Fig. 5b and Supplementary Fig. 9c), demonstrating the *in vivo* tumor targeting ability of RT11-i and TMab4-i."

Comment #3-10) In their xenografts experiments it would be good to see effect of the RT11-i in for example NRas mutant tumours and compare it with KRas.

Response: As the reviewer recommended, we assessed the *in vivo* anti-tumor efficacy of RT11-i in nude mice harboring pre-established xenografts of oncogenic NRas^{Q61K} mutant-harboring HT1080 human soft tissue sarcoma. In the same manner as KRas mutant tumors, antibodies were intravenously dosed at 20 mg/kg every 2 days. RT11-i inhibited the growth of HT1080 tumor xenografts, showing approximately 70% more tumor-growth inhibition (TGI) and approximately 63% greater reduction in tumor weight, compared to those after treatment with TMab4-i. As a result, the *in vivo* sensitivity of HT1080 (~70% TGI) was greater than that of KRas mutant-carrying SW480 (~56% TGI) and LoVo (~46% TGI) tumors. However, with these limited cases, it is difficult to state that NRas-mutant tumors are more sensitive to RT11-i than KRas-mutant tumors. At this point, we speculate that the *in vivo* sensitivity of the tumors to RT11-i indicates their differential dependency on oncogenic Ras signaling.

[Fig. 6a] and its legend in the revised manuscript,

"(a) *In vivo* anti-tumor efficacy of RT11-i compared to that of vehicle and TMab4-i controls, analyzed by measuring the tumor volume during treatment of female BALB/c nude mice harboring the indicated tumor xenografts. Antibodies were intravenously dosed at 20 mg/kg every 2 d (indicated by the arrows). Error bars, \pm s.d. ($n = 8$ per group)."

In the **[Results]** (pp. 12-13) of the revised manuscript,

"We next assessed the *in vivo* anti-tumor efficacy of RT11-i through intravenous

injection into mice harboring pre-established oncogenic Ras mutant tumor xenografts (SW480 (KRas^{G12V}), LoVo (KRas^{G13D}), and HT1080 (NRas^{Q61K})), or Ras^{WT} tumors (Colo320DM). Compared to the PBS-treated vehicle control, TMab4-i slightly retarded the growth of SW480 and HT1080 tumors, which was accompanied by reduced phosphorylation of Akt (Fig. 6a,b and Supplementary Fig. 10 and 11); this could be attributed to the anti-tumor activity of the RGD10 moiety, through blocking integrin $\alpha\beta3$ - and/or $\alpha\beta5$ -mediated tumor angiogenesis and growth^{39,40}. Nonetheless, the growth of LoVo tumors was not affected by TMab4-i. Importantly, RT11-i markedly inhibited the growth of the three oncogenic KRas mutant tumor xenografts, showing approximately 46 – 70% more tumor-growth inhibition (TGI) (Fig. 6a) and approximately 42 – 63% greater reduction in tumor weight, compared to those after treatment with TMab4-i (Supplementary Fig. 10b). However, in case of Ras^{WT} Colo320DM tumors, no significant difference was observed in anti-tumor efficacy between RT11-i and TMab4-i. During the antibody treatments the mice did not exhibit any significant body weight loss (Supplementary Fig. 10c). Thus, the additional anti-tumor activity of RT11-i for Ras mutant tumors could be ascribed to the blocking activity of oncogenic Ras signaling. The *in vivo* sensitivity of the tumors to RT11-i was in the following order: HT1080 (~70% TGI) > SW480 (~56% TGI) > LoVo (~46% TGI), which roughly correlated with *in vitro* colony formation assay results (Fig. 4f), indicating their different dependency on oncogenic Ras signaling."

In the [Discussion] (pp. 16-17) of the revised manuscript,

"However, RT11-i as a single agent exhibited measurable anti-tumor activity via the oncogenic Ras-specific blocking mechanism, showing a ~46–70% more TGI, when compared to treatment with TMab4-i, in mice harboring oncogenic KRas mutant SW480 and LoVo tumors as well as NRas mutant HT1080 tumors, but not in those harboring Ras^{WT} tumors (Fig. 6), without systemic toxicity (Supplementary Fig. 10c)."

We also performed IHC experiments on the excised tumor tissues of HT1080 xenografts, similar to that performed with SW480 and LoVo tumors, shown in Fig. 6b and Supplementary Fig. 11. The results yielded essentially equivalent results to those of SW480 and LoVo tumors. We are sorry for not showing all of these revised

results using HT1080 xenografts. Please refer to revised Supplementary Fig. 11 in the revised manuscript.

Comment #3-11) In figure 6c authors claim only RT11-i co-localized with KRas at the inner membrane. This is not possible to see on the provided images. As before, higher magnification images should be shown.

Response: Thanks for your comment. We revised the Fig. 6c by showing enlarged areas to clearly demonstrate the co-localization of RT11-i with activated Ras around the inner plasma membrane of cryosectioned tumor tissues.

[Fig. 6c] and its legend in the revised manuscript:

"In (c), the areas in the white boxes are shown at a higher magnification for better visualization. The arrows indicate the co-localization of RT11-i with activated Ras. Scale bar, 10 μ m."

We also revised [Supplementary Fig. 11a] in the same manner to clearly demonstrate the co-localization of RT11-i with activated Ras around the inner plasma

membrane of cryosectioned tumor tissues in the revised manuscript.

Comment #3-12) In discussion authors claim that integrin-bound RT antibody "... did not compromise the inherent biochemical and biological properties of RT11". I don't agree with this since localization and proliferation seems to be different. Also the integrin signal itself has effect on phosphorylation of downstream targets and also on some xenografts. This should be discussed.

Response: We thank you for the insightful comment. In response to the reviewer's comment, we deleted the original sentence and added the following discussion to the revised the manuscript.

In the **[Discussion]** (pp. 16-17) of the revised manuscript:

" RT11-i exhibited selective binding to the cell surface-expressed integrin $\alpha\beta3/\alpha\beta5$ while retaining the binding specificity of RT11 to active Ras after cellular internalization. Importantly, when intravenously administered, RT11-i demonstrated reasonable serum half-life and preferential accumulation in tumor tissue (Fig. 5). RT11-i might exert some anti-tumor activity due to the RGD10-mediated integrin blocking activity^{39,40}, particularly for integrin $\alpha\beta3/\alpha\beta5$ -overexpressing tumor cells, based on the effects of TMab4-i on SW480 and HT1080 xenograft tumors (Fig. 6a,b). However, RT11-i as a single agent exhibited measurable anti-tumor activity via the oncogenic Ras-specific blocking mechanism, showing a ~46–70% more TGI, when compared to treatment with TMab4-i, in mice harboring oncogenic KRas mutant SW480 and LoVo tumors as well as NRas mutant HT1080 tumors, but not in those harboring Ras^{WT} tumors (Fig. 6), without systemic toxicity (Supplementary Fig. 10c)."

Minor points) On page 18, on the first paragraph: "Importantly, RT11-i markedly inhibited the tumor growth of the three tumor types carrying oncogenic KRas..." Authors present only two tumours carrying KRas mutations. This should be corrected.

Response: Thank you for your correction. We corrected the typos in the revised manuscript. Please refer to our response to comment #3-10.

Again, we greatly appreciate all of your great comments.

REVIEWERS' COMMENTS:

Reviewer #1 (Remarks to the Author):

I had a very favorable opinion of the MS in the first review cycle and my comments were rather minor. The authors have addressed all my comments and I feel that the current version of the MS is suitable for publication in NC

Reviewer #2 (Remarks to the Author):

The authors have done a commendable job in responding to my comments on the first draft of the manuscript. I feel that the current draft is acceptable for publication.

Reviewer #3 (Remarks to the Author):

In this reviewed version of the paper Shin and colleagues have add significant new data to their original manuscript that have strength their findings.

All the questions raised in my first revision of the paper have been addressed satisfactory and I think the paper will add new values to the field.